# Remote ischemic conditioning counteracts the intestinal damage of necrotizing enterocolitis by improving intestinal microcirculation

Yuhki Koike[1,2,3,14], Bo Li [1,2,14], Niloofar Ganji[1,2,14], Haitao Zhu[1,2,14], Hiromu Miyake[1,2], Yong Chen[1,2], Carol Lee[1,2], Maarten Janssen Lok[1,2], Carlos Zozaya [4], Ethan Lau[1,2], Dorothy Lee[1,2], Sinobol Chusilp [1,2], Zhen Zhang[1,2], Masaya Yamoto[1,2], Richard Y. Wu[5,6], Mikihiro Inoue[3], Keiichi Uchida[3], Masato Kusunoki[3], Paul Delgado-Olguin [1,7,8], Luc Mertens[9], Alan Daneman[10], Simon Eaton [11], Philip M. Sherman [5,12] & Agostino Pierro [1,2,13✉]

Necrotizing enterocolitis (NEC) is a devastating disease of premature infants with high mortality rate, indicating the need for precision treatment. NEC is characterized by intestinal inflammation and ischemia, as well derangements in intestinal microcirculation. Remote ischemic conditioning (RIC) has emerged as a promising tool in protecting distant organs against ischemia-induced damage. However, the effectiveness of RIC against NEC is unknown. To address this gap, we aimed to determine the efficacy and mechanism of action of RIC in experimental NEC. NEC was induced in mouse pups between postnatal day (P) 5 and 9. RIC was applied through intermittent occlusion of hind limb blood flow. RIC, when administered in the early stages of disease progression, decreases intestinal injury and prolongs survival. The mechanism of action of RIC involves increasing intestinal perfusion through vasodilation mediated by nitric oxide and hydrogen sulfide. RIC is a viable and non-invasive treatment strategy for NEC.

[1] Translational Medicine Program, The Hospital for Sick Children, Toronto, ON, Canada. [2] Division of General and Thoracic Surgery, Translational Medicine, The Hospital for Sick Children, Toronto, ON, Canada. [3] Departments of Gastrointestinal and Pediatric Surgery, Mie University Graduate School of Medicine, Tsu, Japan. [4] Division of Neonatology, The Hospital for Sick Children, Toronto, ON, Canada. [5] Cell Biology Program, Research Institute, Division of Gastroenterology, Hepatology and Nutrition, Hospital for Sick Children, Toronto, ON, Canada. [6] Department of Laboratory Medicine and Pathobiology, Faculty of Medicine, University of Toronto, Toronto, ON, Canada. [7] Department of Molecular Genetics, University of Toronto, Toronto, ON, Canada. [8] Heart & Stroke Richard Lewar Centre of Excellence, Toronto, ON, Canada. [9] The Labatt Family Heart Center, Cardiology, The Hospital for Sick Children, University of Toronto, Toronto, ON, Canada. [10] Department of Diagnostic Imaging, Division of Nuclear Medicine, The Hospital for Sick Children, University of Toronto, Toronto, ON, Canada. [11] UCL Great Ormond Street Institute of Child Health, London, UK. [12] Faculty of Dentistry, University of Toronto, Toronto, ON, Canada. [13] Department of Surgery, University of Toronto, Toronto, ON, Canada. [14] These authors contributed equally: Yuhki Koike, Bo Li, Niloofar Ganji, Haitao Zhu. ✉email: agostino.pierro@sickkids.ca

Necrotizing enterocolitis (NEC) is a devastating intestinal disease that affects 5–7% of preterm infants and remains a major unsolved clinical challenge in neonatology[1]. Gut immaturity in neonates with NEC can lead to intestinal inflammation, progressing in severe cases to necrosis and perforation. NEC results in high mortality (30–50% in advanced cases)[2], neurodevelopmental impairment[3,4], intestinal failure[5], reduced quality of life, and high treatment costs estimated at $500M to $1B per year in the USA[2]. Although mortality of premature infants continues to decrease, mortality due to NEC remains high[6]. Present treatments are only supportive; they include antibiotics, bowel rest, and surgery to remove necrotic bowel if necessary. There have been few innovations in either medical or surgical management of NEC other than general improvements in neonatal care. Altogether, there is a need for devising a treatment strategy to avoid the progression from initial inflammatory changes to more advanced intestinal injury in NEC.

Unfortunately, prevention and management of NEC are challenging as the etiology of this disease remains incompletely understood. NEC is considered multifactorial, with several risk factors proposed including prematurity, formula feeding, microbial colonization, inflammation, and hypoxia-ischemia[7,8]. Enteral feeding in the early stage of life can lead to intestinal ischemia[7,9]. This stems from the fact that the immature intestine has an inadequate intestinal hemodynamic response to feeding which prevents the intestine from fulfilling its oxygen demand after feeding[10]. Formula feeding in the early neonatal period (postnatal day P5) did not stimulate an increase in intestinal perfusion, contrary to the response observed later at P9[10]. Consistently, there is emerging evidence in the literature suggesting that derangements in microcirculatory intestinal blood flow are associated with experimental NEC and play a significant role in this disease[11–14]. Therefore, modulating the immature intestinal microcirculation could prove to be a viable strategy to counteract the feeding-induced hypoxia and prevent the development of NEC. We have previously demonstrated that this can be done by promoting vasodilation in the microvasculature mediated by nitric oxide (NO)[10]. NO belongs to the subfamily of endogenous gaseous signaling molecules, similar to hydrogen sulfide ($H_2S$). Findings by other studies have also supported a role for NO and $H_2S$ to protect the intestine against the development of NEC by increasing microvascular blood flow via vasodilation[15,16]. These findings suggest that promoting sufficient intestinal blood flow in the immature intestine could prevent mucosal damage and improve the outcome of NEC.

Remote ischemic conditioning (RIC) is a therapeutic strategy for protection of tissue/organs against the detrimental effects of ischemia. RIC involves application of brief cycles of ischemia and reperfusion to a remote tissue/organ in order to protects distant tissue/organs from sustained ischemic damage. The phenomenon of ischemic preconditioning was first described in the canine heart wherein brief cycles of ischemia and reperfusion of the coronary artery reduced myocardial infarction size[17]. RIC was also found to confer cardio-protective effects elicited from applying cycles of non-lethal ischemia-reperfusion to organs remote from the heart[18,19]. Further experiments revealed that RIC protects other organs in addition to the heart from acute ischemic injury[20]. In the stomach for instance, RIC stimulus to the heart or liver significantly reduced gastric mucosal injury, improved gastric blood flow, and suppressed plasma pro-inflammatory cytokine levels in rat model of gastric ischemic injury[21].

We performed the current study to initially establish whether human neonates with NEC present mucosal hypoxia and impairment of microvasculature in the intestine. Subsequently, we evaluated if RIC was effective in counteracting NEC damage and altering intestinal perfusion. In addition, we explored the RIC mechanism of action. Our findings suggest that RIC promotes vasodilation of the microvasculature in the immature intestine via endogenous vasodilators such as NO and $H_2S$, preserves intestinal perfusion, reduces mucosal hypoxia, and ultimately improves the outcome of NEC.

## Results

**Intestinal microcirculation is compromised in human NEC.** The intestinal damage in NEC is affecting most commonly the terminal ileum. To evaluate whether the disease is characterized by villi hypoxia and impaired intestinal microcirculation, we have studied the terminal ileum resected from human infants with NEC. We compared, within the same individuals, different portions of the ileum at varying distance from the most damaged area. In addition, we compared the NEC ileum with that excised from age-matched newborn infants without NEC, operated for congenital intestinal obstructions. The most affected terminal ileum, closest to the site of necrosis and perforation, had severe damage in the top of the villi demonstrated by the highest and most extensive expression of the hypoxia marker (Hypoxia-inducible factor 1α, HIF1α)[22] compared to the less affected and not-affected ileum (Fig. 1a–c). Remarkably, in the same damaged villi, the microvascular vessels were completely absent. This was indicated by lower expression of vascular endothelial marker (cluster of differentiation 31, CD31), compared to the less affected and not-affected ileum (Fig. 1a–c). Hence, hypoxia and microvascular derangements are maximal in the most affected intestinal area of NEC. Conversely, ileum farther away from this most affected area regains normal features resembling tissues from non-NEC control neonates (Fig. 1b, c).

These findings indicate that NEC is associated with mucosal hypoxia, reduced number of endothelial cells and reduced microvascular vessels, suggestive of compromised intestinal perfusion.

**RIC improves NEC-induced intestinal damage.** To counteract the above derangements in intestinal perfusion and mucosal hypoxia in infants affected by NEC, we have performed pre-clinical studies to evaluate the effects of RIC using a well-established experimental model of NEC. First, to determine the optimal timing of RIC during NEC induction, we studied the extent of intestinal injury from P5 to P9. We noted progressive increase in intestinal injury, with the most significant morphological changes and inflammation starting at P7 (Fig. 2a and Supplementary Fig. 1a). Induction of NEC, resulted in separation of submucosa and lamina propria, partial loss of villi, and villus sloughing as well as increased inflammation at P9 (Fig. 2b and Supplementary Fig. 1b). Next, to address whether RIC differentially impacts the progression of NEC based on timing, RIC was given at P5 when intestinal damage was minimal or not yet present (Stage 1 RIC), at P6 when changes in intestinal damage were first detected (Stage 2 RIC), or at P7 (Stage 3 RIC) when significant changes in intestinal injury had occurred (Fig. 2c). Interestingly, following application of Stage 1 or 2 RIC, there was recovery of intestinal villi and separation of the submucosa and lamina propria was significantly reduced (Fig. 2d, e). Intestinal inflammation was also attenuated following application of Stage 1 or 2 RIC (Supplementary Fig. 1b). Most importantly, both Stages 1 and 2 RIC significantly improved survival of pups with NEC (Fig. 2f). In contrast, Stage 3 RIC did not provide any significant protection on intestinal injury (Fig. 2d, e) and inflammation (Supplementary Fig. 1b) and did not enhance survival (Fig. 2f). Therefore, all subsequent experiments focused on Stages 1 and 2 RIC only. To investigate whether the RIC-mediated protection of

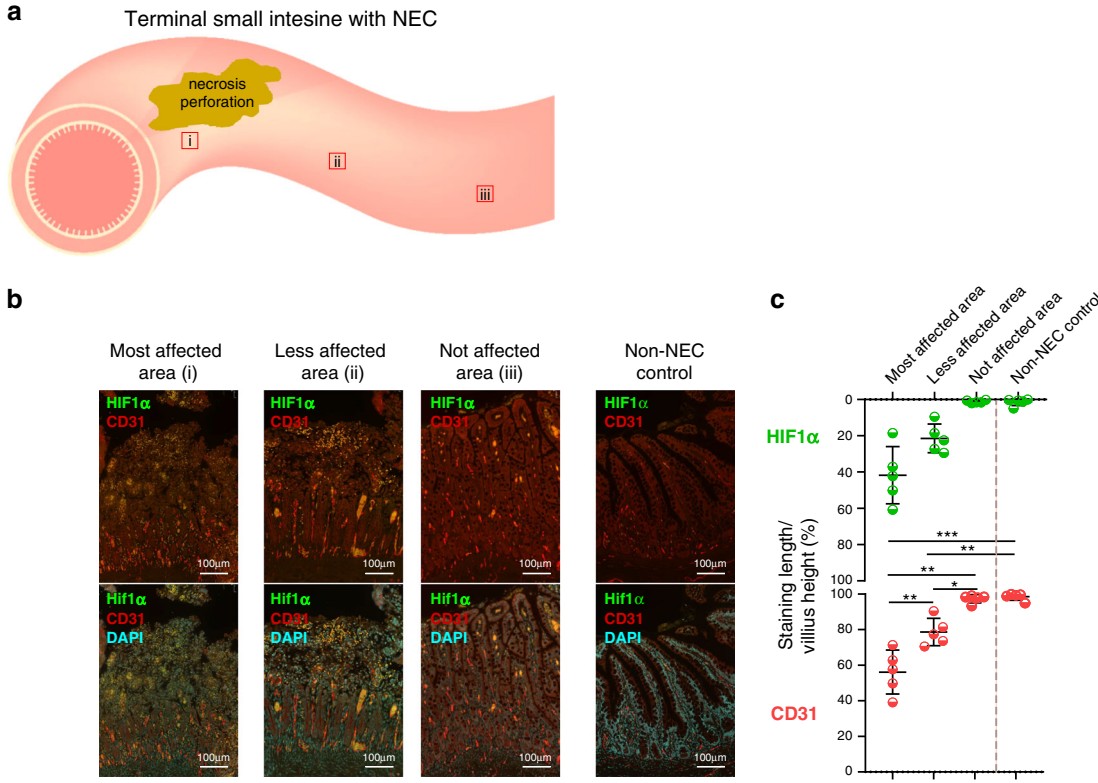

**Fig. 1 Human NEC is associated with mucosal hypoxia and impairment of the microvasculature. a** The terminal ileum of NEC patients ($n = 5$) and non-NEC control patients ($n = 5$) were analyzed and compared using immunofluorescence staining. **b, c** The intestine was divided into three regions: (i) the most affected ileum close to the area of necrosis and perforation ($n = 5$), (ii) the less affected ileum distant from the necrosis/perforation ($n = 5$), and (iii) the not affected ileum farther away from the damaged area ($n = 5$). The most affected ileum had the highest expression of hypoxia-inducible factor 1α (HIF1α) and the lowest expression of endothelial cell marker (CD31), compared to less affected, not affected areas and non-NEC controls. There was a progressive decrease in HIF1α expression and progressive increase in CD31 expression from NEC-damaged to less damaged area. Immunofluorescence staining and imaging and analysis of slides were performed by blinded investigators and data werr compared using two-sided one-way ANOVA with post hoc Turkey test and repeated measure for the comparison of different areas within the NEC patient and no pairing comparison to the non-NEC control (*$p$ < 0.05, **$p$ < 0.01, ***$p$ < 0.001). Scale bars are equivalent to 100 μm in all the images shown, and data are presented as mean ± SEM. Experiments were repeated independently 3 times, with similar results. Source data are provided as a Source Data file.

the intestine in NEC is dependent on the endothelium, we investigated the effects of RIC in *eNOS* knock out pups. Stages 1 and 2 RIC did not improve intestinal morphology, reduce inflammation, or extend survival in *eNOS* knock out pups (Fig. 2g, h and Supplementary Fig. 1c). To confirm that this is not due to underlying difference in basal injury between *eNOS* and wildtype mice, we also compared the inflammation and histology in non-NEC control *eNOS* pups, which had shown comparable levels of injury (Fig. 2g, h and Supplementary Fig. 1c).

Taken together, our findings indicate that RIC-mediated intestinal protection in NEC is dependent on endothelium-mediated vasodilation.

**RIC preserves intestinal microcirculation.** Given that *eNOS* plays a role in the RIC-mediated protection against intestinal damage of NEC, next we assessed the downstream effect of RIC-induced *eNOS* signaling on mesenteric circulation. To do this, we quantified perfusion across the intestinal wall using Doppler ultrasound daily during NEC progression from P5 to P9[23,24]. Intestinal wall perfusion was calculated as average flow velocity (mm/s) of multiple abdominal regions (Fig. 3a). In agreement with intestinal damage, intestinal wall flow velocity was significantly reduced in NEC pups, compared to breastfed controls. Remarkably, both Stages 1 and 2 RIC increased flow velocity in the intestinal wall, indicating improved intestinal perfusion (Fig. 3b–d and Supplementary Movies 1–4).

We then investigated whether the protective effect of RIC extends deeper into the intestinal microcirculation using live imaging by two photon laser scanning microscopy or TPLSM (Fig. 4a). Using this technique, we have previously demonstrated that mouse pups in the early neonatal period (P5) failed to increase microcirculatory perfusion in the submucosa in response to a single gavage formula feeding[10]. This is contrary to the response observed in the pups later at P9. These findings suggest that the immature intestinal microcirculatory system does not respond to feeding, leading to the development of NEC. However, administration of RIC to P5 pups counteracted the poor microcirculatory response to formula feeding and preserved the arteriole flow velocity, diameter, and flow volume in the intestine (Supplementary Fig. 2a–c). More significantly, NEC led to impaired submucosal perfusion, but this was preserved in pups subjected to Stage 1 or 2 RIC (Supplementary Movies 5–8). Quantification of blood flow in real time demonstrated that arteriole flow velocity, arteriole diameter, and flow volume were compromised at P9 in NEC pups, but were improved by Stages 1 and 2 RIC (Fig. 4b–d). Thus, RIC improves perfusion across the entire intestinal wall and in the intestinal submucosa. A preserved intestinal microcirculation predicts reduced ischemic damage in the intestinal villi[25]. To test this contention, we investigated the extent of intestinal ischemia by staining with the hypoxia marker pimonidazole. NEC was associated with increased ischemia at the tip of the villi. However, ischemia was restored to normal levels in NEC pups

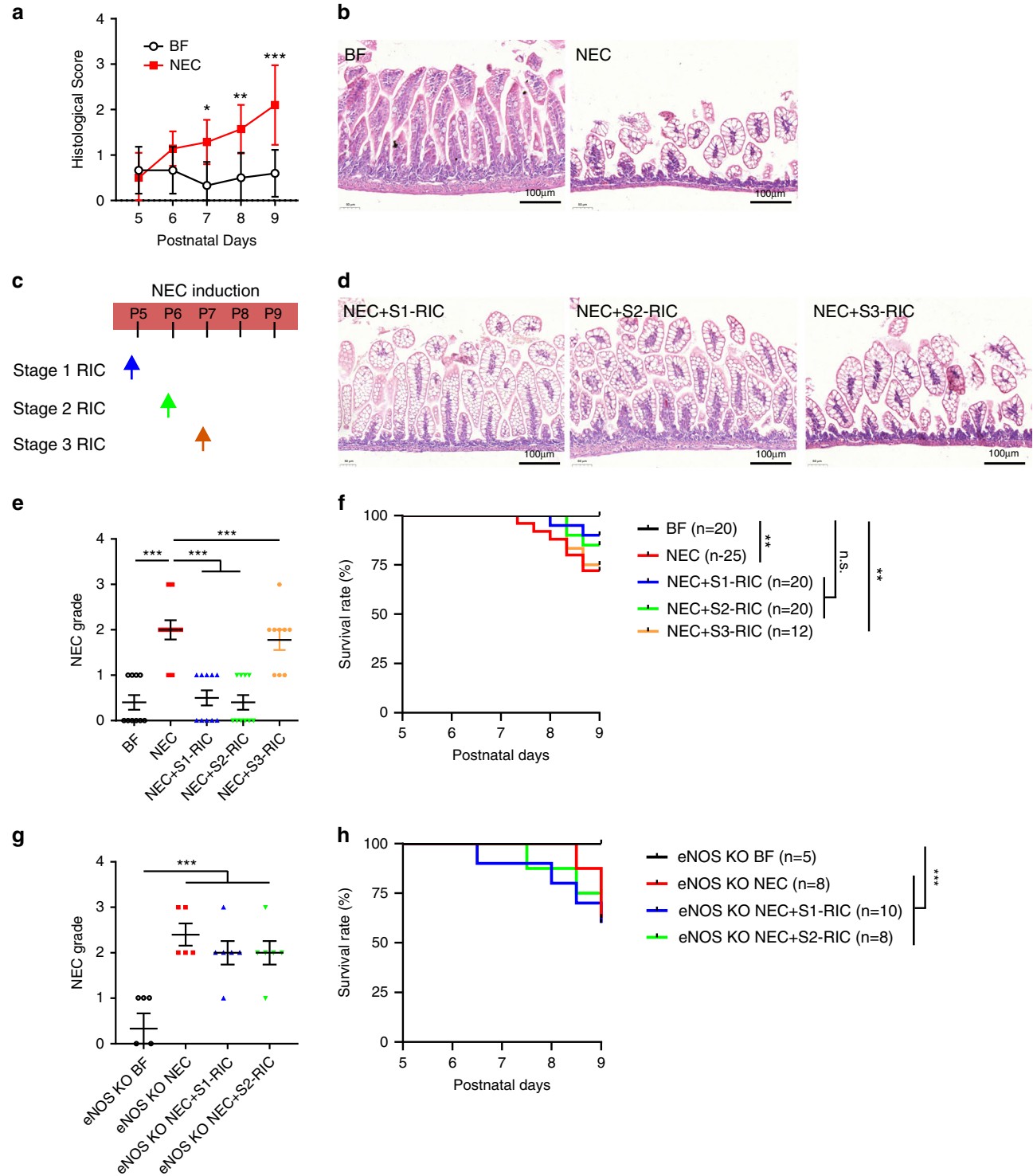

receiving Stage 1 or 2 RIC (Fig. 5a, b), which agrees with improved intra-villi perfusion. Consistently, there was a marked constriction and decrease in density (Fig. 5c, d) and a decrease in the length of intra-villi arterioles during NEC (Fig. 5e, f). However, both Stages 1 and 2 RIC resulted in recovery of the diameter, density, and height of intra-villi arterioles (Fig. 5c–f). These results indicate that RIC preserves integrity of the villi microvasculature in NEC. Accordingly, necrosis in the intestinal epithelium, revealed by staining with Sytox Green[26], was increased in NEC, but significantly reduced by Stages 1 and 2 RIC (Fig. 5g, h; Supplementary Movies 9–12).

These results suggest that RIC confers protective effects by preserving intestinal microcirculation during NEC.

**RIC is dependent on vasodilatory signaling mediators**. To characterize the signaling molecules that mediate RIC-induced NEC protection, next we evaluated the roles of gasotransmitters NO and $H_2S$ in the ileum during NEC development. To test NO involvement, we measured the effect of Stage 1 or 2 RIC on intestinal wall flow velocity in *eNOS* knockout pups subjected to NEC. Baseline intestinal wall velocity of *eNOS* knockout mice was

**Fig. 2 RIC improves intestinal damage during experimental NEC. a** To establish the timing of RIC regimen in the initial development of NEC, intestinal injury was investigated in C57BL/6 mouse pups from P5 to P9 (BF P5–P8: $n = 6$; BF P9: $n = 10$; NEC P5: $n = 6$; NEC P6–8: $n = 7$; NEC P9: $n = 10$). **b** Morphology of the ileum was compared between breastfed (BF) control ($n = 10$) and NEC pups ($n = 10$) using hematoxylin and eosin staining. Scale bars are equivalent to 100 μm in the images shown. Experiments were repeated independently 3 times, with similar results. **c** RIC was given in three distinct time points, each consisting of two episodes of RIC that are 48 h apart: RIC was initiated on P5 (Stage 1, $n = 10$), P6 (Stage 2, $n = 10$), or P7 (Stage 3 RIC, $n = 10$). **d** Effect of Stages 1–3 RIC on intestinal morphology were compared in NEC pups at P9. Scale bars are equivalent to 100 μm in the images shown. **e** Histological slides were graded for BF ($n = 8$), NEC ($n = 10$), NEC + Stage 1 RIC ($n = 10$), NEC + Stage 2 RIC ($n = 10$), and NEC + Stage 3 RIC ($n = 10$) by 3 investigators blinded to treatment allocation based on the NEC histopathological scoring system. Mice with histological grade ≥2 were considered to have NEC. **f** Stages 1 and 2 RIC equally enhanced survival of NEC pups up to sacrifice on P9. Stage 3 RIC did not improve NEC survival. **g** Effect of Stages 1 and 2 RIC on intestinal morphology in *eNOS* knockout NEC pups (*eNOS* BF: $n = 5$, *eNOS* NEC: $n = 5$, *eNOS* NEC + Stage 1 RIC: $n = 6$, *eNOS* NEC + Stage 2 RIC: $n = 6$). Histological slides were graded by 3 blinded investigators as described above. **h** *eNOS* knockout pups with NEC receiving Stage 1 or 2 RIC had a similar survival rate to *eNOS* knockout pups with NEC alone. Data were compared using two-sided one-way ANOVA with post hoc Turkey test (*$p < 0.05$; **$p < 0.01$, ***$p < 0.001$). Data are presented as mean ± SEM. Source data are provided as a Source Data file.

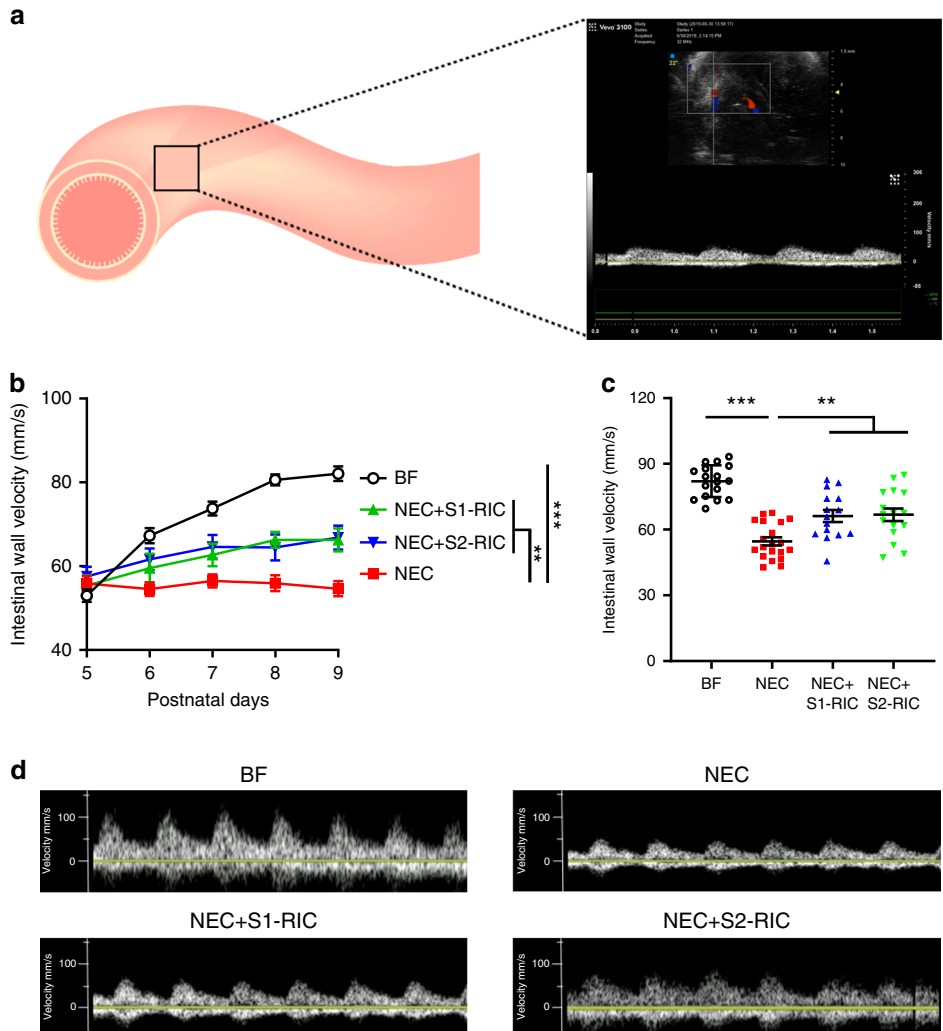

**Fig. 3 RIC improves intestinal wall perfusion during NEC development. a** Intestinal wall perfusion was measured using Doppler ultrasound and calculated as average flow velocity (mm/s) of multiple abdominal regions. **b** Intestinal wall perfusion measured daily from P5 to P9 showed reduced perfusion in NEC pups which was improved following conditioning with Stage 1 or 2 RIC ($n = 6$ per group; minimum of 3 readings of different abdominal quadrants were obtained per pup; **$p < 0.01$; ***$p < 0.001$). **c** Intestinal wall perfusion measured on P9 in NEC pups showed reduced perfusion compared to breastfed (BF) control, which was preserved in NEC pups receiving Stage 1 or 2 RIC ($n = 6$ per group; minimum of 2 readings of different abdominal quadrants were obtained per pup; **$p < 0.01$; ***$p < 0.001$). **d** Representative images of arterial waves obtained with the Doppler ultrasound in P9 pups from listed groups. Data were compared using two-sided one-way ANOVA with post hoc Turkey test and data are presented as mean ± SEM. Source data are provided as a Source Data file.

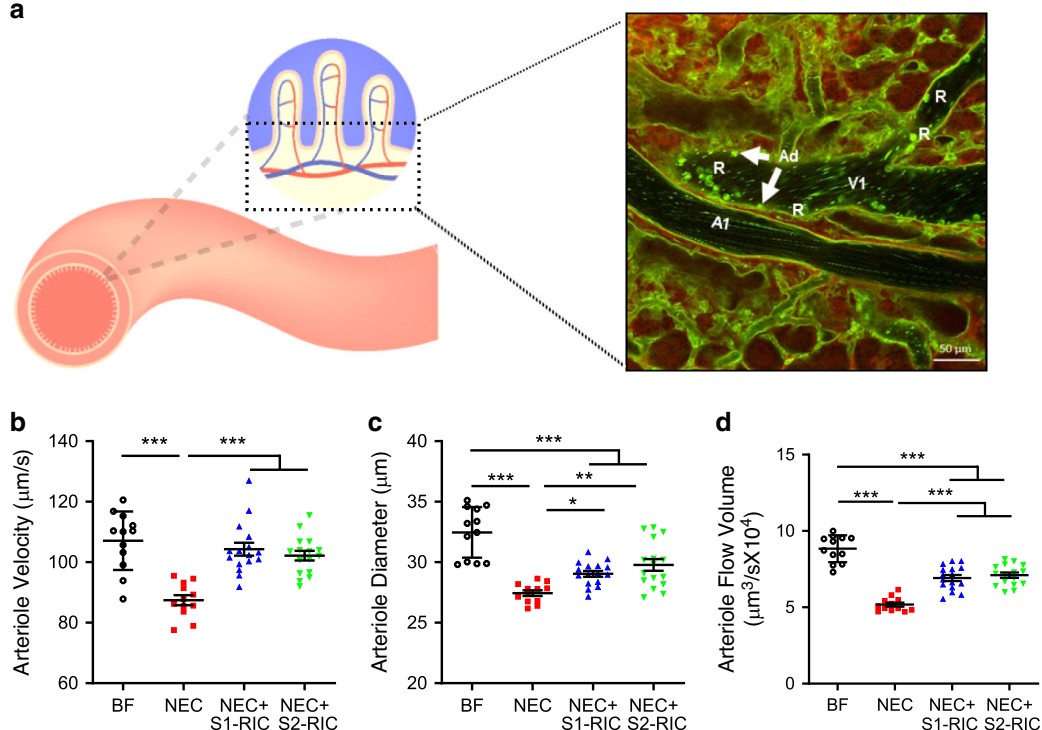

**Fig. 4 RIC preserves intestinal microcirculation in the submucosa. a** Depiction of submucosal arterioles (A1) and submucosal venules (V1) recorded via in vivo live imaging by TPLSM. Experiments were repeated independently 3 times, with similar results. Arteriole response in the ileal submucosa at P9 in terms of arteriole **b** blood flow velocity (μm/s), **c** diameter (μm), and **d** flow volume [(μm)$^3$/s] compared between breastfed (BF) control, NEC, NEC with Stage 1 RIC, and NEC with Stage 2 RIC groups (BF and NEC: $n = 6$; NEC + Stage 1 or 2 RIC: $n = 8$; minimum of 2 readings were obtained per group). Data were compared using two-sided one-way ANOVA with post hoc Turkey test ($*p < 0.05$; $**p < 0.01$; $****p < 0.0001$). Data are presented as mean ± SEM. Source data are provided as a Source Data file.

markedly reduced compared to wild type mice indicating tenuous perfusion of the intestine (Supplementary Fig. 3); hence measurements of intestinal perfusion during NEC and RIC in *eNOS* knockout mice were not reliable. Therefore, the role of NO in the improvements in intestinal perfusion induced by RIC was tested by administering the inhibitor of NO synthase, L-NAME. Stages 1 and 2 RIC-mediated increase of intestinal blood flow dynamics was abolished in NEC pups treated with L-NAME (Fig. 6a, b, d, e, g, h). This demonstrates that NO inhibition via L-NAME or genetic knockout abolishes the protection from RIC.

H$_2$S was previously shown to promotes vasodilation, increase microvascular blood flow in the intestine, and improve the injury from NEC[27,28]. To test its involvement in the RIC-mediated protection, we evaluated the levels of the H$_2$S-synthesizing enzyme CBS in the ileum. Compared to breastfed controls or NEC alone, administration of Stage 1 or 2 RIC increased expression of CBS in the ileum (Supplementary Fig. 4a, b). Such an increase was prevented by administration of inhibitors of H$_2$S-synthesizing enzymes (Supplementary Fig. 4a, b). We hypothesized that the beneficial effects of RIC on intestinal perfusion are dependent on H$_2$S-mediated vasodilation. Indeed, Doppler ultrasound showed that both Stages 1 and 2 RIC-mediated preservation of intestinal wall perfusion and flow velocity were abolished upon administration of inhibitors of H$_2$S-synthesizing enzymes (Supplementary Movies 13 and 14). Quantification of flow velocity by Doppler ultrasound revealed that Stage 1 or 2 RIC did not preserve flow velocity in the intestinal wall when inhibitors of H$_2$S-synthesizing enzymes were given (Fig. 6a, b). Similarly, real-time assessment by TPLSM indicated that RIC failed to preserve submucosal perfusion upon administration of H$_2$S-synthesizing enzyme inhibitors (Supplementary Movies 15

and 16). Analysis by TPLSM revealed that H$_2$S-synthesizing enzyme inhibitors abolished Stages 1 and 2 RIC-mediated preservation of arteriole velocity (Fig. 6d, e) and flow volume (Fig. 6g, h). Administration of an exogenous H$_2$S donor (NaHS) had protective effects like Stage 1 or 2 RIC on intestinal perfusion. Remarkably, such protective effect was abolished following delivery of H$_2$S-synthesizing enzyme inhibitors (Fig. 6c, f, i). This suggests that even during NEC induction, some endogenous H$_2$S synthesizing activity remains. As expected, arteriole diameter was reduced during NEC and increased in response to RIC, but not when H$_2$S and NO synthesis inhibitors were given in addition to RIC (Supplementary Fig. 5). Collectively, our data reveals that the mechanism of action of RIC in counteracting impaired intestinal perfusion during NEC is due to vasodilation mediated by NO and H$_2$S.

**RIC preservation of perfusion is vital to improve NEC outcome.** RIC significantly preserves intestinal morphology and perfusion, and thus might be a viable strategy to improve NEC outcome. Administration of NO synthase inhibitor and H$_2$S-synthesizing enzyme inhibitors blunted the beneficial effects of Stages 1 and 2 RIC in improving intestinal injury and inflammation during NEC (Fig. 7a–e and Supplementary Fig. 6a, b). In contrast, treatment with the H$_2$S donor, NaHS, decreased intestinal injury and inflammation similar to RIC; however, this effect was blunted in the presence of H$_2$S-synthesizing enzyme inhibitors (Fig. 7f, g and Supplementary Fig. 6c).

To test whether the RIC-mediated improvements of intestinal morphology and blood flow influence the final outcome of NEC, we investigated whether RIC could improve survival of

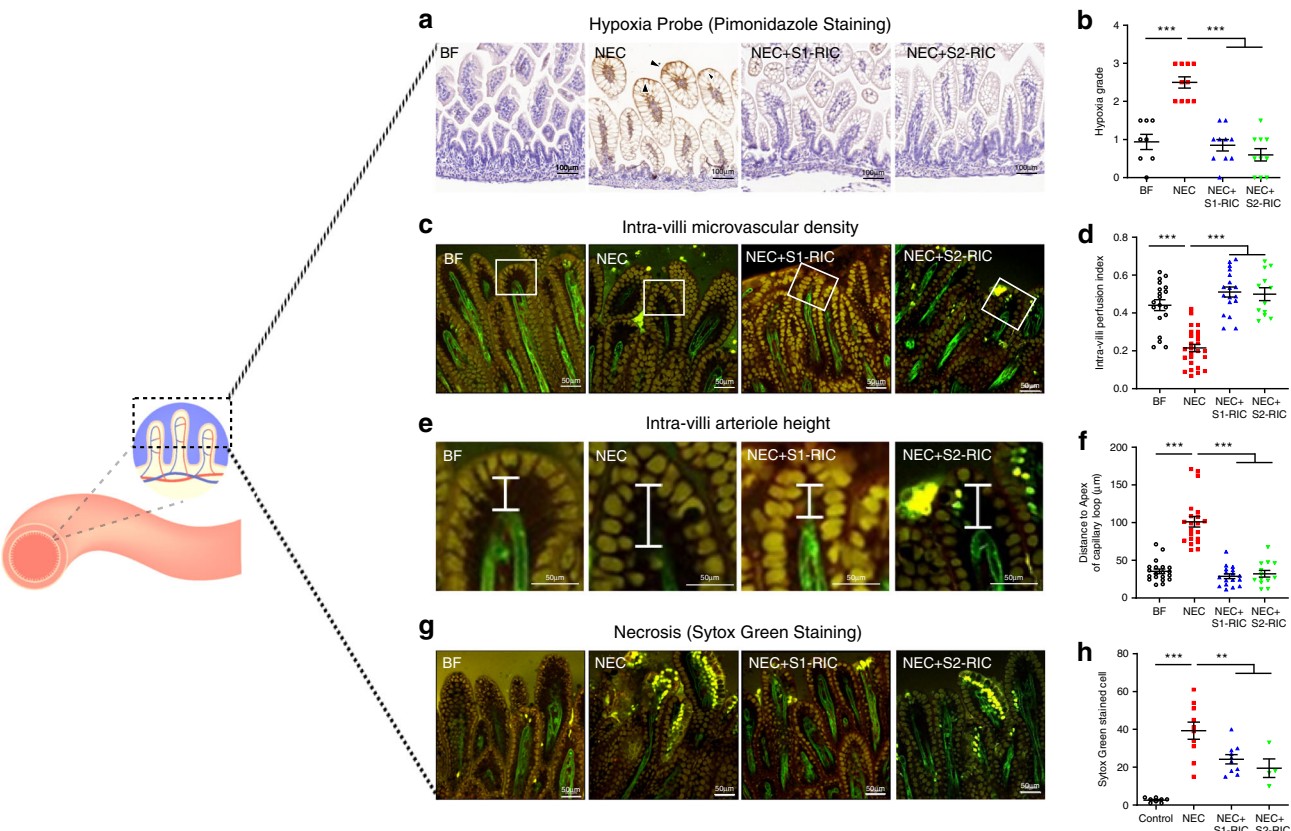

**Fig. 5 RIC improves intestinal villi microvasculature and reduces ischemia and necrosis of enterocytes at the villi tip. a, b** Pups were injected with pimonidazole, a sensitive marker which allows localization of intestinal ischemia. Data represent immunohistochemistry of pimonidazole in the ileum comparing breastfed (BF) control ($n = 8$), NEC ($n = 10$), NEC with Stage 1 RIC ($n = 10$), and NEC with Stage 2 RIC ($n = 10$). Scale bars are equivalent to 100 µm in the images shown. **c** Intra-villi microvasculature was investigated in $Rosa^{mT/mG/+}$;*Tie2-Cre* using TPLSM and compared between BF, NEC, NEC with Stage 1 RIC, and NEC with Stage 2 RIC ($n = 4$ per group; and minimum of 2 measurements were obtained per group). **d** Villus perfusion index was calculated as the ratio of the area of intra-villi arterioles to the area of the whole villi and compared between the listed groups. **e, f** The distance between the apex of the capillary loop and the apical side of the villi epithelium was measured and compared between listed groups. **g** Cell death (yellow staining in the tip of villi) in the ileal epithelium was detected using Sytox Green, a marker of necrosis which does not permeate into live cells but binds to cellular nucleic acids in dead cells, staining them with intense green fluorescence and was compared between listed groups. **h** Necrotic cells were counted in the listed groups (*$p < 0.05$; **$p < 0.01$; ****$p < 0.0001$). Data were compared using two-sided one-way ANOVA with post hoc Turkey test (**$p < 0.01$; ****$p < 0.0001$). Scale bars in (**c**, **e** and **g** are equivalent to 50 µm, and data are presented as mean ± SEM. Source data are provided as a Source Data file.

experimental NEC. To characterize the outcome of NEC when RIC is applied, we quantified survival rate beyond P9. Pups from all groups except breastfed controls continued to receive gavage feeding three times per day and were monitored until endpoint. Strikingly, compared to NEC alone, survival was significantly longer after either Stage 1 or 2 RIC (Fig. 8a). Furthermore, we found that administration of $H_2S$-synthesizing enzyme inhibitors abrogated protection against NEC by Stage 1 or 2 RIC, resulting in mortality similar to NEC alone (Fig. 8b). These findings provide evidence that the RIC improves intestinal perfusion via $H_2S$ to maintain tissue integrity and counteract the effects of NEC.

Finally, to assess whether improved intestinal perfusion is sufficient to improve the outcome of NEC, we investigated the effects of two nonspecific vasodilators, papaverine[29] and captopril[30]. Both vasodilators improved intestinal injury (Supplementary Fig. 7a), reduced inflammation (Supplementary Fig. 7b), and enhanced survival (Supplementary Fig. 7c). To confirm the significance of vasodilation-mediated improvement of intestinal perfusion in the mechanism of action of RIC, we assessed the effect of an intestinal vasoconstrictor methoxamine[31] in the presence of RIC. Administration of methoxamine abolished the beneficial effect of Stage 1 or 2 RIC in prolonging survival (Supplementary Fig. 7d).

These findings provide further evidence that the protective effects of RIC are due to changes in intestinal perfusion.

**RIC is a safe maneuver**. We demonstrated that Stage 1 or 2 RIC in breastfed control pups did not alter intestinal morphology, inflammation, and intestinal perfusion (Supplementary Fig. 8a–f). Similarly, in these animals, administration of $H_2S$-synthesizing enzyme inhibitors did not alter the above parameters (Supplementary Fig. 8a–f).

To further validate the safety and feasibility of RIC, we used three neonatal motor tests[32] and compared the outcomes between P9 breastfed control pups and P9 breastfed pups conditioned with Stage 1 or 2 RIC. The hind limb foot angle test was performed by measuring the foot angle using a line drawn from the mid-heel through the middle (longest) digit (Supplementary Fig. 9a). Conditioning with Stage 1 or 2 RIC did not produce any abnormalities in the hind limb foot angle. In order to assess the right/left hind limb strength and neuromuscular function, we used the hind limb suspension test. Conditioning with Stage 1 or 2 RIC did not produce any significant difference in the hind limb suspension score (Supplementary Fig. 9b). The surface righting test was performed in order to assess the motor ability of mouse

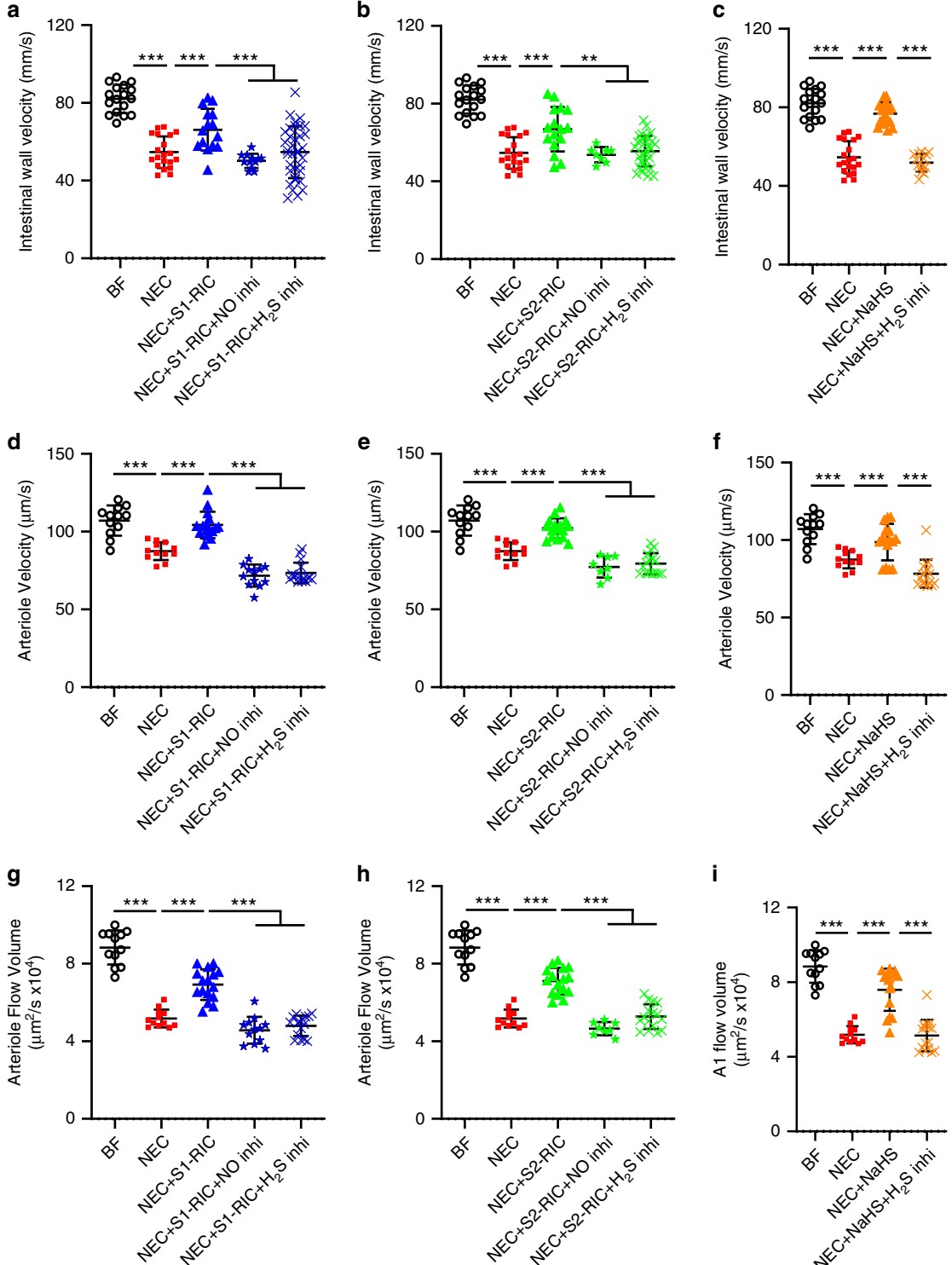

**Fig. 6 RIC preserves intestinal microcirculation via endogenous vasodilators nitric oxide and hydrogen sulfide.** Administration of NO-synthase inhibitor and $H_2S$-synthesizing enzyme inhibitors abolished **a** Stage 1 RIC and **b** Stage 2 RIC-mediated preservation of intestinal wall flow velocity (mm/s) measured with Doppler ultrasound. **c** Treatment with NaHS, exogenous donor of $H_2S$, improved intestinal wall flow velocity (mm/s), but not in the presence of $H_2S$-synthesizing enzyme inhibitors. Administration of NO-synthase inhibitor and $H_2S$-synthesizing enzyme inhibitors abolished the **d**, **g** Stage 1 RIC- and **e**, **h** Stage 2 RIC-mediated preservation of submucosal arteriole velocity (μm/s) and arteriole flow volume [(μm)³/s] respectively, measured using TPLSM in real time. **f**, **i** Treatment with NaHS improved submucosal arteriole velocity (μm/s) and arteriole flow volume [(μm)³/s] respectively, but not following administration of $H_2S$-synthesizing enzyme inhibitors. Data were compared using two-sided one-way ANOVA with post hoc Turkey test ($n = 8$ per group; minimum of 2 readings were obtained per group; **$p < 0.01$; ***$p < 0.001$). Data are presented as mean ± SEM. Source data are provided as a Source Data file.

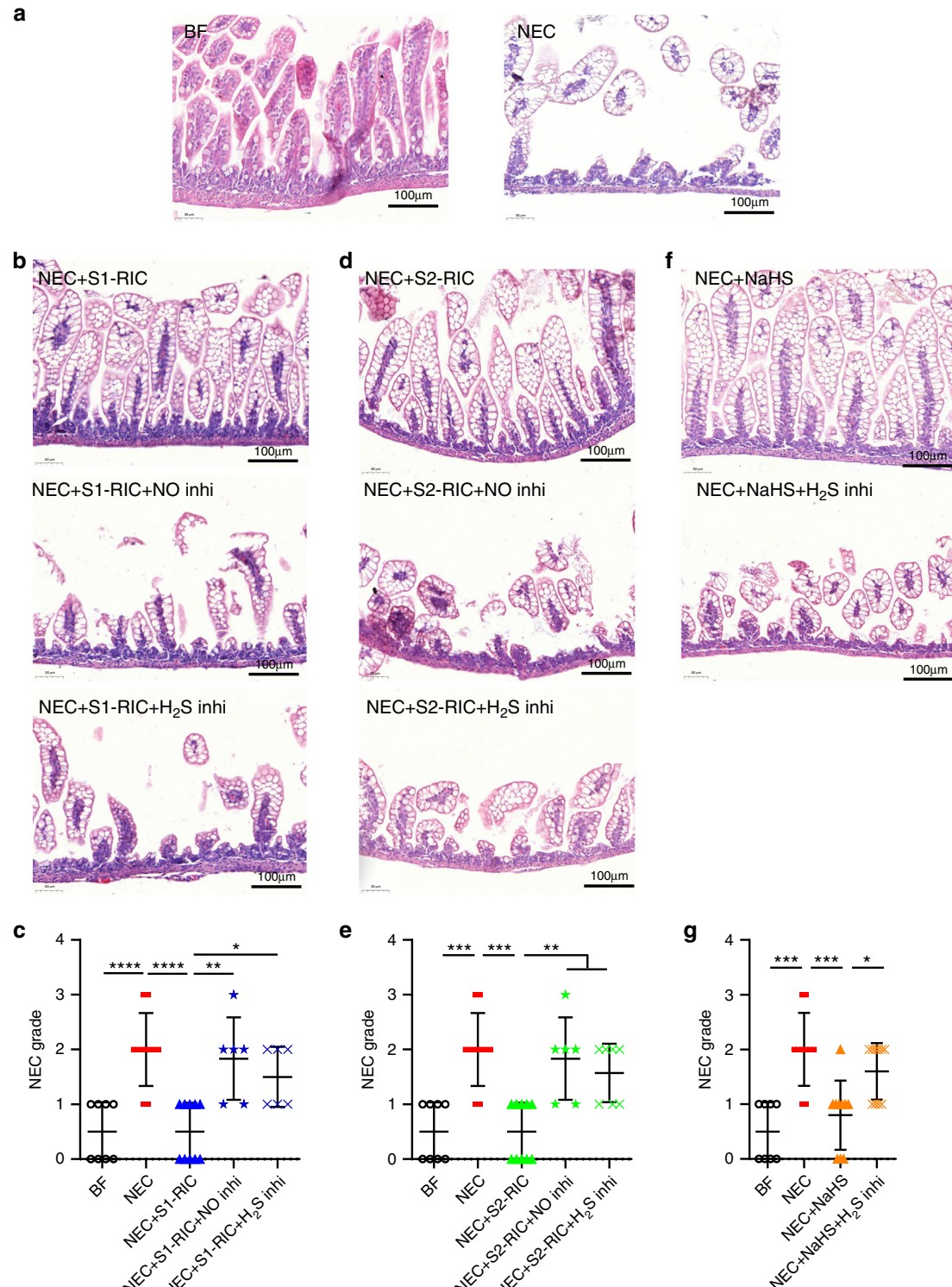

**Fig. 7 RIC-mediated preservation of intestinal perfusion via nitric oxide and hydrogen sulfide is required to improve intestinal injury during NEC.**
**a** Morphology of the ileum was assessed using hematoxylin and eosin staining in breastfed (BF) ($n = 8$) control and NEC pups ($n = 10$). Intestinal injury increased in NEC pups receiving **b**, **c** Stage 1 RIC following administration of inhibitors of NO-synthase ($n = 6$) or $H_2S$-synthesizing enzymes (n = 6), as well as in pups receiving **d**, **e** Stage 2 RIC following administration of inhibitors of NO-synthase (n = 6) or $H_2S$-synthesizing enzymes (n = 7) compared to NEC pups receiving Stage 1 RIC ($n = 10$) or Stage 2 RIC ($n = 10$) without drug treatment. Morphology of the ileum was assessed using hematoxylin and eosin staining and histological slides were graded by 3 investigators blinded to treatment allocation based on the NEC histopathological scoring system that defines mice with NEC grade ≥ 2 as NEC positive. **f**, **g** Treatment with NaHS improved intestinal injury in NEC pups, resulting in a lower NEC grade, but not following treatment with $H_2S$-synthesizing enzyme inhibitors (BF: $n = 8$; NEC: $n = 10$; NEC + NaHS: $n = 10$, NEC + NaHS+$H_2S$-synthesizing enzyme inhibitors: $n = 10$). Data were compared using two-sided one-way ANOVA with post hoc Turkey test (*$p < 0.05$; **$p < 0.01$; ***$p < 0.001$). Scale bars are equivalent to 100 μm in **a**, **b**, **d** and **f**. Data are presented as mean ± SEM. Source data are provided as a Source Data file.

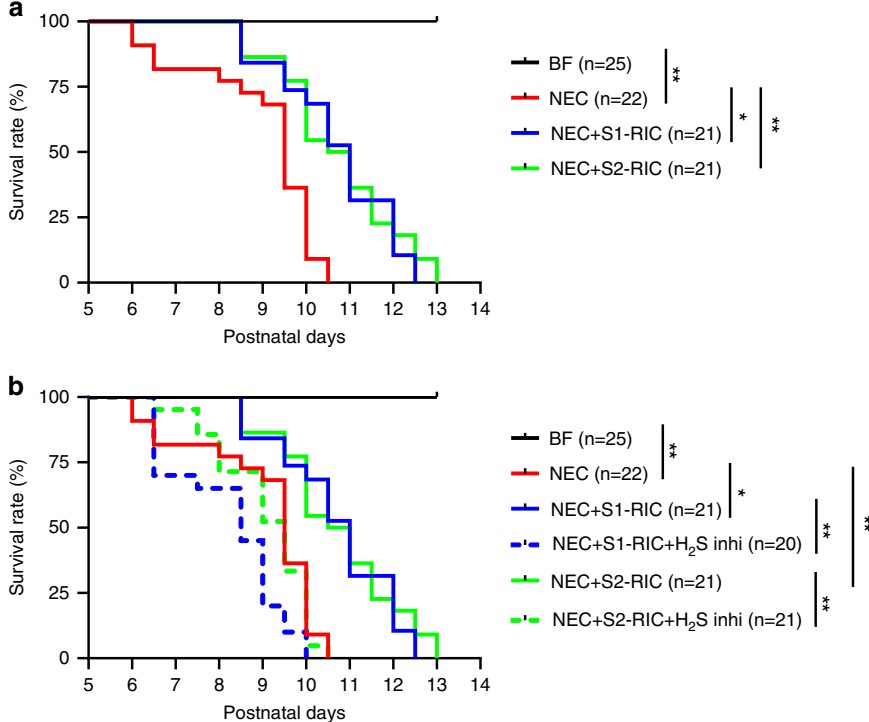

**Fig. 8 RIC enhanced survival during and after NEC and this effect was abolished following administration of H$_2$S-synthesizing enzyme inhibitors.**
**a** After P9, pups from all groups except breastfed (BF) controls, continued to receive gavage feeding three times per day by single investigator blinded to treatment allocation and were monitored until death occurred. Pups receiving Stage 1 or 2 RIC survived significantly longer after NEC induction (*$p < 0.05$; **$p < 0.01$). **b** Survival rate after P9 was similar between NEC pups and NEC pups receiving Stage 1 or 2 RIC and given H$_2$S-synthesizing enzyme inhibitors (*$p < 0.05$; **$p < 0.01$). Survival curves were compared using the logrank test (*$p < 0.05$; **$p < 0.01$). Data are presented as mean ± SEM. Source data are provided as a Source Data file.

pups to turn over onto their feet from the supine position. There was no significant difference in the time to turn over in pups conditioned with Stage 1 or 2 RIC (Supplementary Fig. 9c; Supplementary Movies 17–19). These results show that Stages 1 and 2 RIC are safe and do not produce any deficits in motor function.

## Discussion

NEC remains the most severe and lethal gastrointestinal neonatal emergency. This devastating disease is characterized by intestinal hypoxia and derangements in intestinal microcirculation. The potential of RIC to protect against the intestinal damage of NEC in preterm infants has not been explored to date. We demonstrate that RIC offers protection against the early stages of experimental NEC by ameliorating intestinal injury and ultimately prolonging survival. RIC conveys protection by preserving perfusion, submucosal microcirculation, and villi microvasculature in the immature intestine (Fig. 9). RIC applied to the hind limb of mouse pups appears to be safe as it did not cause intestinal injury and had no effect on the function or mobility of the limb.

Previous experimental studies in the heart and the brain have demonstrated that RIC targets microcirculation in the distant ischemic target organ[33–36]. RIC enhanced collateral circulation in a murine model of focal cerebral ischemia[33], elevated cerebral blood flow in the rat ischemic brain during stroke recovery[34], and improved coronary collateral circulation in a rabbit model of myocardial ischemia[35]. These reports suggest a potential protection to organs via improved circulation. However, to date, the role of RIC on neonatal homeostasis and intestinal perfusion in NEC has not been explored. We were excited to find in our study that conditioning with RIC was able to counteract the poor response

to feeding in the early neonatal period by improving intestinal microcirculation. Our results suggest that by promoting vasodilation, RIC improved microvascular dynamics in the intestine and thereby protected the intestine from NEC.

This study confirms that microvascular blood flow to the injured intestine is compromised during NEC[8,11–13,37]. Excitingly, we demonstrated that RIC improved intestinal perfusion during NEC in the entire intestinal wall. Using TPLSM to investigate real-time intestinal microcirculation in vivo, we obtained further insight into the mechanism of action of RIC. We discovered that RIC reverses the derangements in microvascular blood flow to the injured intestine and reconstitutes a normal submucosal blood flow. We also demonstrated that NEC induction caused a significant reduction in diameter and height of the arterioles perfusing the villi, consistent with our observation of increased hypoxia in the tip of the villi. However, Stage 1 or 2 RIC reversed these changes, indicating improved diameter and height of intra-villi arterioles as well as reduced hypoxia in the villi tip. NEC is normally associated with villus core separation, and in more severe cases, sloughing of the villi and presence of necrotic tissue at the tip of the villi[25,37,38]. We confirmed increased necrotic enterocytes following ischemia at the tip of the villi during NEC. This can be explained by our in vivo observations that microvascular perfusion at the tip of the villi was markedly reduced. Conditioning with RIC decreased ischemia and necrosis of enterocytes at the villi tip. This study shows that RIC improves intra-villi microvascular perfusion in the intestine with NEC and reduces necrosis at the villi tip.

To explore the role of vasodilation and enhanced intestinal perfusion in the protective effects of RIC, we used chemical inhibitors of NO and H$_2$S, potent gaseous vasodilators in the developing gastrointestinal tract[16,27,28,39]. We demonstrated that

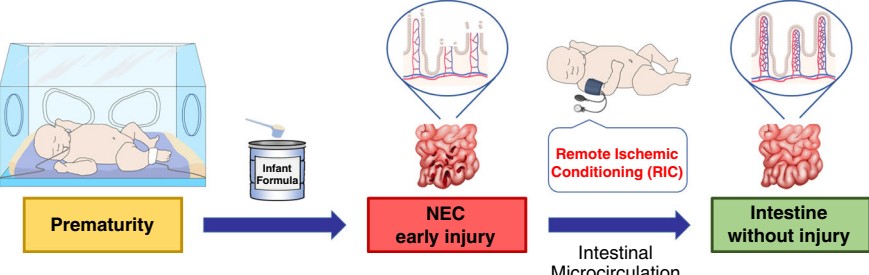

**Fig. 9 Remote ischemic conditioning counteracts the intestinal damage of necrotizing enterocolitis (NEC) by improving intestinal microcirculation.** Prematurity and formula feeding are among the main risk factors contributing to the development of NEC, a devastating disease of premature infants characterized by intestinal inflammation and ischemia. NEC is characterized by derangements in intestinal microcirculatory blood flow, villus core separation, sloughing of the villi, and presence of necrotic tissue at the villi. Remote ischemic conditioning is a therapeutic maneuver whereby application of brief cycles of ischemia and reperfusion to a limb protects a distant organ from sustained ischemic damage. During the initial stage of experimental NEC, remote ischemic conditioning improves intestinal injury, reduces inflammation, and enhances survival. The mechanism of action of remote ischemic conditioning involves preservation of intestinal microcirculation that is mediated by the vasodilators, hydrogen sulfide and nitric oxide. Remote ischemic conditioning is a noninvasive treatment strategy for neonatal NEC.

inhibition of synthesis of NO abolished the RIC-mediated improvements in intestinal perfusion and resulted in poor intestinal morphology and increased inflammation. $H_2S$ is known to improve intestinal perfusion via an *eNOS*-dependent mechanism[16], suggesting that intestinal microcirculation is controlled by $H_2S$ upstream of the NO pathway. Inhibitors of $H_2S$ synthesizing enzymes CBS and CSE are commonly used to inhibit endogenous $H_2S$ synthesis, and hence to investigate the biological effects of suppressing $H_2S$-mediated vasodilation[40]. In our study, inhibition of CBS together with CSE abolished the RIC-mediated improvements in intestinal microcirculation and resulted in NEC despite RIC as demonstrated by increased intestinal injury, loss of intestinal villi and submucosal integrity, increased intestinal inflammation, and ultimately poor survival. Conversely, administration of NaHS, an $H_2S$ donor, significantly reduced the severity of NEC and provided a similar protective effect to RIC, but this effect was abolished following administration of $H_2S$ synthesis inhibitors. This finding is consistent with previous reports in other diseases demonstrating that the beneficial effects of exogenous $H_2S$ require synthesis of endogenous $H_2S$[41–43]. Future studies should examine whether the protective effects of exogenous $H_2S$ result from a direct action or an indirect effect through modulating endogenous $H_2S$ synthesis and signaling.

NO and $H_2S$ have anti-inflammatory actions independent of their vasomotor effects, which could also contribute to the protection mediated by RIC. Thus, it is possible that the chemical inhibitors of NO and $H_2S$ block not just the vasomotor effects, but also other biochemical and physiological effects. We also used *eNOS* knockout mice in order to isolate the effects of RIC on the endothelium. We demonstrated that the beneficial action of RIC was abolished in these pups, ultimately resulting in poor survival. However, it has been reported the expression of *eNOS* in cell types other than the endothelium[44]. Previous study demonstrated that *eNOS* is expressed in macrophages and may play a role in the initiation of inflammation[45]. It is still possible that the lack of protection by RIC in *eNOS* KO mice may be due to the role of *eNOS* in modulating inflammatory responses. Hence, we cannot conclude that the mechanism of action of RIC is exclusively targeted to the endothelium and regulation of blood flow.

Additionally, we demonstrated that administration of papaverine and captopril, two potent vasodilatory mediators, resulted in improved intestinal injury, reduced inflammation, and enhanced survival. These findings were consistent with our previous finding that captopril supplementation of formula reduces the severity of intestinal damage and the incidence of NEC, by eliciting intestinal vasodilation and enhancing mesenteric blood

flow[30]. Conversely, administration of the vasoconstrictor methoxamine to NEC pups with Stage 1 or 2 RIC abolished the protective effects of RIC and resulted in worse survival rates than NEC alone. Taken together, our findings suggest that the mechanism of action of RIC involves improving the intestinal perfusion during NEC by promoting vasodilation. The attraction of RIC is related to its simplicity, experimental effectiveness, and avoidance of the administration of compounds which can cause undesirable side effects.

Different models of RIC have been used to protect distant organs against ischemic injury[40,46,47]. We used intermittent occlusion of blood flow to the hind limb via a tourniquet as it was previously proven to be effective in causing ischemia[48]. Previous studies have suggested that RIC activates two distinct time frames of protection against ischemia reperfusion injury of the brain and heart[49–51]. The initial window of protection occurs immediately after the RIC stimulus and lasts for 2 h whereas the second window of protection occurs 12–24 h after the RIC stimulus and lasts 48–72 h[49,50]. Moreover, it has been reported that compared with a single episode of RIC, repeated episodes were more protective in reducing oxidative stress, lipid peroxidation, and inflammation in the ischemic myocardium[51]. Based on such findings, we chose three distinct time points of RIC, each consisting of two episodes of RIC that are 48 h apart, through the course of disease progression in our model. To take into account the clinical presentation of infants with NEC, and the potential clinical translation of RIC, we studied the effects of Stages 1–3 RIC. Stage 1 RIC was initiated on P5, in the presence of minimal intestinal damage to simulate human Stage I NEC[52–54]. Stage 2 RIC was initiated during NEC development on P6 when intestinal damage was more evident, to simulate Stage II NEC which is characterized by moderate disease[52–54]. Finally, Stage 3 RIC was given in later stages of NEC development on P7 when advanced intestinal damage was present, to emulate advanced Stage III NEC[52–54]. We have shown that Stages 1 and 2 RIC are equally effective in mitigating the intestinal damage caused by NEC and most importantly enhancing survival during and after NEC induction. In contrast, Stage 3 RIC did not confer protection against intestinal damage of NEC, suggesting that RIC is no longer effective when ischemic injury in the intestine has become advanced. Our findings suggest that RIC is only effective in counteracting the derangements in microvascular blood flow in the immature intestine and improving the outcome of NEC when administered in early stages of disease progression. This indicates that RIC can become a therapeutic strategy for the management of initial stages of NEC avoiding further disease progression.

Clinical trials have been performed in adults[55–57] and in children[58–62] which suggest benefits from RIC protecting various organs including heart, lung, and kidney. In addition, a systematic review and meta-analysis evaluating randomized trials, found that compared with controls, RIC significantly reduced the recurrence of stroke or transient ischemic attacks[63]. However, the advantage of RIC on specific human organs remains controversial as large randomized controlled trials showed no improvement after myocardial infarction[64] or in relation to cardiac surgery[65,66]. Only two trials in adults have focused on the effects of RIC on the intestine[12,67]. One trial demonstrated benefit after abdominal aortic aneurism repair, after which intestinal ischemia/reperfusion-induced injury is expected[12]. In contrast, the other trial found no intestinal changes after cardiopulmonary bypass which can cause moderate and transient intestinal injury[67]. Previous studies in children undergoing cardiac surgery have indicated that RIC can be applied in infants and children without deleterious effects[68]. However, to the best of our knowledge, RIC has never been given to small preterm neonates with NEC. Although RIC is a simple and attractive maneuver, there is a need to evaluate feasibility and safety in this patient population.

We have not observed any change in intestinal morphology, inflammation, and intestinal perfusion by RIC on breastfed control pups. We further validated the safety of RIC via three neonatal motor tests[32]. No deterioration in motor function was detected in response to RIC. These experimental observations have strong implications for translation into the clinical setting. In neonates, RIC can be applied noninvasively by brief cycles of inflating and deflating a standard blood pressure cuff on the upper arm or leg[69].

Despite significant medical advancements, the prevention and treatment of NEC remain important challenges in neonatology. Multicenter studies reported that: (1) the median time between initial clinical concern and confirmed medical NEC is 31 h and for advanced disease requiring surgery is 57 h[70], and (2) the majority of patients that develop advanced surgical NEC do not present with a fulminant course, resulting in disease progression[71]. In addition, intestinal reparative changes are seen in 68% of infants with an acute first episode of NEC, suggesting that NEC can be an evolving process, with intestinal damage occurring days before requiring an operation[72]. RIC may prevent progression of the disease and avoid irreversible intestinal damage. RIC is a noninvasive, safe, feasible, and easy-to-use physical maneuver, which has been used to protect brain and heart in clinical trials[73,74]. Finally, RIC presents a strategy for harnessing the body's endogenous protection against the intestinal injury incurred by ischemia during NEC, with the advantage that the RIC stimulus can be applied to a remote organ, distant from the intestine. Taken together, RIC can be implemented in the clinic, with the potential to minimize the need for aggressive surgical intervention in the treatment of NEC. Our next step is to investigate the feasibility and safety of RIC in premature neonates before embarking on a clinical trial designed to demonstrate its efficacy.

In summary, our data provide evidence that RIC promotes recovery from NEC by improving intestinal morphology, reducing intestinal inflammation, and ultimately enhancing survival. In the current study, we have demonstrated that the mechanism of action of RIC involves preserving intestinal perfusion by promoting vasodilation and enhancing microvascular blood flow in the immature intestine. The full mechanism of action of RIC is not yet entirely elucidated as it may also involve regulation of inflammation. Our discovery indicates that RIC can become a therapeutic option for the initial stages of NEC and its remarkable benefits can modify the way the management of this devastating disease is conducted worldwide.

## Methods

In accordance with best practice, all experiments conducted in this paper were performed in triplicate and were repeated for three separate iterations without reusing experimental samples. Assessment of histological scoring, immunofluorescence, Doppler ultrasound, TPLSM, and survival was conducted by investigators blind to the allocation of treatment.

**Human small intestine.** Ethical approval for this study was obtained from the Research Ethics Board of the Hospital for Sick Children, Toronto, Canada (protocol #1000056881). All methods performed in the study were carried out in accordance with the approved guidelines and regulations. Informed consent was obtained from the Legally Authorized Representatives of the infants. Tissue analysis was done with approval from the Hospital for Sick Children and in accordance with anatomical tissue procurement guidelines. Except for the study principal investigator (A.P.) and study coordinator, no study personnel analyzing these samples had access to personal identification information.

Samples were obtained from the terminal ileum of infants with NEC stored in the Division of Pathology. The ileum was resected during emergency laparotomy for acute active NEC ($n = 5$). Age-matched control samples ($n = 5$) were obtained from resected ileum of infants undergoing surgery for less-severe diseases of the intestine (Hirschsprung's disease, meconium ileus). All infants were premature and were operated on during the first 7 weeks of life. Immunostaining was carried out for Hif1-α (mouse monoclonal, Novus NB100-105) (1:250) and CD31 (rabbit polyclonal Abcam ab18364) (1:100) using same methodology described in the mice experiments above.

**Animals.** All animal experiments were approved by the Animal Care Committee of the Hospital for Sick Children, Toronto, Canada (no. 32238), and were conducted in accordance with the guidelines and regulations. Experimental pups (mixed gender) undergoing NEC induction were separated from their mothers at postnatal day 5 (P5) while controls remained with mothers to breastfeed. Separated pups (body weight: 3–5 g) were housed in incubators maintained at 37 °C, on a 12:12 h light/dark cycle, and with beddings provided. NEC was induced in neonatal C57BL/6 mice or in eNOS knockout mice via gavage feeding of hyperosmolar formula, hypoxia, and oral administration of lipopolysaccharide (4 mg/kg), from P5 to P9 as described previously[75]. Breastfed littermates served as controls. On P9, pups were sacrificed, and terminal ileum was harvested for analysis. $Rosa^{mT/mG/+}$; Tie2-Cre or $Rosa^{GFP}$ mice were used for in vivo analysis with TPLSM[76]. To further characterize the outcome after RIC, we measured the survival rate beyond P9. Pups from all groups except breastfed controls continued to receive gavage feeding three times per day, without intravenous fluid and were monitored until end-point.

RIC was administered to neonatal NEC-induced mice via intermittent occlusion of arterial flow in the left hind limb using a tourniquet—a surgical rubber vessel loop of 1 mm diameter. The RIC stimulus was given in four cycles, each consisting of 5 min occlusion followed by 5 min reperfusion. Hind limb occlusion and reperfusion in each cycle of RIC were verified by monitoring the change in color of the limb (pale pink following occlusion and red following reperfusion), which was consistently achieved within <30 s. However, due to technical limitations for mouse pups of this size (postnatal day 5–9 and body weight 3–5 g), we could not obtain mechanical measurements of pressure. Based on previous studies demonstrating that RIC is most effective for 48 h[49,50] and when given repeatedly[51], each RIC stimulus was given in two non-consecutive days 48 h apart. Stage 1 RIC was initiated at P5 when intestinal damage was minimal or not yet present, mimicking human Stage I NEC[52–54]. Stage 2 RIC was initiated at P6 when changes in intestinal damage became detectable, mimicking human Stage II NEC[52–54]. Stage 3 RIC was initiated at P7 when intestinal injury and inflammation were more advanced, mimicking human Stage III NEC[52–54].

To validate the presence of an ischemic environment in NEC pups, 60 mg/kg of the hypoxia marker, pimonidazole (Hypoxyprobe Inc., MA, USA) were injected intraperitoneally at P9 before the last feed[10]. Mice were euthanized 90 min after feeding and routine methods of immunostaining were performed.

**Intestinal vasomotor agents.** To define the role of NO-mediated vasodilation in the mechanism of action of RIC, NEC was induced in enzyme nitric oxide synthase (eNOS) homozygous knockout mice. In addition, NEC pups receiving Stage 1 or 2 RIC were given NO synthase inhibitor, Nω-Nitro-L-arginine methyl ester hydrochloride (L-NAME) (60 mg/kg/day, Sigma-Aldrich), which was added to the formula prior to gavage feeding[77].

H$_2$S is also a gaseous vasodilator and its levels cannot be directly measured at tissue level. The principal pathway for synthesis of H$_2$S in mammals involves the enzymatic conversion of the amino acid substrate, L-cysteine to pyruvate, ammonium, and H$_2$S via cystathionine-β-synthase (CBS) and cystathionine-γ-lyase (CSE)[78]. We used inhibitors of H$_2$S in order to assess whether the protection conveyed by RIC depends on H$_2$S-mediated vasodilation. For this purpose, we evaluated in NEC pups, undergoing Stage 1 or 2 RIC, the effects of AOAA (Boc-aminooxy-acetic acid), inhibitor of CBS, and PAG (DL-Propargylglycine), inhibitor of CSE, both administered daily from P5 to P9 by gavage (10 mg/kg/day). To assess whether exogenous H$_2$S administration can mimic the effect on intestinal

microcirculation and the protection conveyed by RIC, we investigated the effect of exogenous NaHS, an $H_2S$ donor (10 mg/kg/day), in NEC pups not undergoing RIC. NEC control groups received phosphate buffered saline as control drug.

To assess whether general vasodilators could mimic the effect of RIC in improving intestinal perfusion and NEC outcome, papaverine (Sigma-Aldrich), and captopril (Sigma-Aldrich) were supplemented with formula at dosages of 20 mg/kg/day and administered to NEC pups.

To assess whether a vasoconstrictor agent would abolish the RIC-mediated improvement in NEC outcome, Methoxamine hydrochloride (Sigma-Aldrich) was supplemented with formula at 150 μg/kg from P5 to P9, and administered to NEC pups receiving Stage 1 or 2 RIC.

**Histological assessment and immunofluorescence staining.** After sacrificing the pups on P9, 1 cm samples of terminal ileal tissue were harvested, fixed in 4% paraformaldehyde, embedded in paraffin, cross-sectioned (5 μm), and counter-stained with hematoxylin and eosin by standard protocols. Histology slides were blindly assessed by three independent investigators using a published scoring system in which confirmed NEC is defined as mice with a grade 2 or above[79].

Sections of terminal ileum were immunostained with 1 in 500 dilutions of primary antibodies for polyclonal rabbit anti-CBS antibody (Proteintech, 14787-1-AP) or pimonidazole (Hypoxyprobe) followed by incubation with 1 in 1000 diluted Alexa Fluor-conjugated secondary antibody (Invitrogen, Carlsbad, California, United States) and DAPI (Vector Laboratories, Burlington, ON) for visualization of cell nuclei. For subsequent reactions, a streptavidin-biotin complex peroxidase kit (LASB + Kit, Dako, Denmark) was used. Slides were analyzed using a Nikon TE-2000 digital microscope equipped with a Hamamatsu C4742-80-12AG camera. Three blinded investigators counted the number of antibody-labeled cells from a minimum of five images. Immunofluorescent images were quantified using ImageJ.

**Gene expression by quantitative PCR.** To measure the effect of RIC on intestinal inflammation, we compared the mRNA expression levels of inflammatory cytokine Interleukin-6 (*IL-6*) in P9 breastfed control pups, NEC-induced pups, and NEC-induced pups conditioned with Stage 1 or 2 RIC. Total RNA (1 μg) was purified from the terminal ileum by Trizol reagent (Invitrogen, Carlsbad, California, United States). Reverse transcription and qRT-PCR were performed[80]. Complimentary DNA (cDNA) was made with qScript cDNA SuperMix (Quantabio, Beverly, Massachusetts, United States) and S1000 Thermal Cycler (Bio-Rad Laboratories, Hercules, California, United States). Real-time PCR was performed with advanced qPCR Master Mix and CFX384 Real-Time System (Bio-Rad Laboratories, Hercules, California, United States). The primer sequences are used for the following primers: *IL-6* (F: CCAATTTCCAATGCTCTCCT; R: ACCACAGTGAGGAATGTCCA), and Glyceraldehyde 3-phosphate dehydrogenase (*Gapdh*, F: TGAAGCAGGCA TCTGAGGG; R: CGAAGGTGGAAGAGTGGGAG). All samples were normalized to the housekeeping gene *Gapdh*.

**Doppler ultrasound measurement of intestinal perfusion.** Doppler ultrasound has been used to study intestinal perfusion both in experiments NEC and humans[23,24]. In this study, we used Doppler ultrasound to measure intestinal perfusion in experimental NEC. Doppler ultrasound of the bowel loops, and gray-scale ultrasound of the peritoneal cavity were performed using VEVO3100 (FUJIFILM VisualSonics, Linear-array probe: 550S, 10–15 Hz) and results were validated by an experienced pediatric radiologist. The abdominal aorta was located using the Gray-scale images and confirmed using the color Doppler sonography mode and then tracked to find the small intestinal wall, with detectable hypoechoic muscle layers at the outside and luminal side of the wall, as well as detectable peristalsis movements. Three to four ultrasound measurements of intestinal wall velocity were performed per animal obtained from the right, left, upper, and lower abdominal quadrants.

**Two photon laser scanning microscopy (TPLSM).** Stereotactically immobilized and anesthetized mice at P9 were analyzed in vivo using an upright microscope (LSM710: Laser-scanning microscope system, Carl Zeiss, Jena, Germany) with ×20 water immersion objective lens (W Plan-Apochromat 20×/1.0 DIC, VIS-IR M27 75 mm). The excitation source (910 nm) was a Mai Tai One Box Ti: Sapphire Laser (Newport Corporation-Spectra-Physics Lasers Division, Mountain View, CA). Two-photon fluorescence signals were collected by an internal detector (non-descanned detection method) at 910 nm. Scan speed was set at 1.27 μs/pixel. The Mai Tai Laser produces light of ~100 fs pulse width (repetition rate, 80 MHz) which was directed onto the sample through the microscope objective, connected to the Zeiss LSM710 microscope. Data were analyzed using ZEN 2.0 lite software (Zeiss, Jena, Germany).

The diameter and blood flow velocity in submucosal arterioles were analyzed and quantified using the line scan technique[10,81]. Arterial blood flow volume was calculated using flow image correlation spectroscopy[6].

The arterial supply to the intestinal villus consists of 1–2 vessels originating from the submucosal vascular network, which ascend centrally to the villus tip[82]. After acquiring images from TPLSM, small pieces of the ileum were resected and cut open for observation of the villus microvasculature[10,83]. Perfusion of the villi was quantified using the villi perfusion index calculated as the average ratio of the area of green-colored intra-villi arterioles to the area of the whole villi. The average distance between the apex of the capillary loop perfusing the villi to the apical side of the villi epithelium within the imaged tissue was measured to assess changes in the height of intra-villi arterioles.

**Safety validation of RIC.** Three neonatal motor tests were performed to investigate outcomes in response to RIC. These included the hind limb foot angle test, the surface righting test, and the hind limb suspension test as described by Feather-Schussler and Ferguson[32]. All tests were repeated three times and outcomes were compared between breastfed control pups and breastfed control pups conditioned with Stage 1 or 2 RIC.

**Statistics.** GraphPad Prism 8 (GraphPad Software, San Diego, California, United States) was used for statistical analyses. All statistics are described in figure legends. Results are presented as mean ± SEM, as data were normally distributed (Kolmogorov-Smirnov test). Survival curves were compared using the logrank test. $p < 0.05$ was considered statistically significant. Groups were compared using Student's *t* test or one-way ANOVA with post-hoc Turkey analysis as appropriate.

**Reporting summary.** Further information on research design is available in the Nature Research Reporting Summary linked to this article.

## Data availability
All data associated with this study are available in the main text or the supplementary materials. Source data are provided with this paper.

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

## Acknowledgements

The authors would like to thank all the members from Dr. Robert Bandsma's and Dr. Elena Comelli's laboratories, as well as Mashriq Alganabi for their helpful comments, suggestions, and graphic design. We thank the Division of Pathology at The Hospital for Sick Children for providing the human samples. The authors kindly acknowledge Laboratory Animal Services of The Hospital for Sick Children for their excellent technical expertise and assistance with animal work. Y.K. was supported by JSPS KAKENHI Grant JP17K11508. B.L. was the recipient of Restracomp Fellowship from The Hospital for Sick Children (HSC). N.G. was the recipient of Restracomp Scholarship from HSC. P.D.O. was supported by the Heart and Stroke Foundation of Canada Grant G-17-0018613, the Natural Sciences and Engineering Research Council of Canada (NSERC) Grant 500865, the Canadian Institutes of Health Research (CIHR) Grant 162208 and PJT-149046, and Operational Funds from HSC. S.E. gratefully acknowledged support from the NIHR Great Ormond Street Hospital Biomedical Research Centre. P.M.S. was supported by the CIHR and by a Canada Research Chair in Gastrointestinal Disease. A.P. was supported by Canadian Institutes of Health Research (CIHR) Foundation Grant 353857 and the Robert M. Filler Chair of Surgery, The Hospital for Sick Children.

## Author contributions

Y.K., B.L., N.G., and H.Z. contributed equally to this paper. Y.K., B.L., N.G., H.Z., H.M., Y.C., C.L., M.J.L., E.L., D.L., S.C., Z.Z., and M.Y. were involved in performing the experiments, acquisition of data, interpretation of data, and writing of the paper. C.Z., R.W., M.I., K.U., M.K., P.D.O., L.M., A.D., and P.M.S. were involved in interpretation of data, and writing of the paper. S.E. was involved in design of the study, interpretation of data, and writing of the paper. A.P. was involved in conceptualization of study hypothesis, design of the study, interpretation of data, writing of the paper, and study supervision.

## Competing interests

The authors declare no competing interests.
