## [Peer Review File · Nature Communications]

Reviewers' Comments:

Reviewer #1:

Remarks to the Author:

Remote ischemic conditioning counteracts the intestinal damage of necrotizing enterocolitis by improving intestinal microcirculation.

Koike et al

The goals of this paper are to determine the effects of remote ischemic conditioning on experimental NEC, and to determine its mechanism of action.

Pups underwent early or late RIC, and noted improved survival and reduced inflammation.

This is an interesting study. I have the following concerns.

1. ischemic preconditioning has been shown to reduce inflammatory responses in many animal models of sepsis and inflammation (see for ex PMID: 30809283, PMID 28437377, PMID 26436208, PMID 24904237, PMID 25037959) among others; these papers offer significant mechanistic insights that could be tested in the current model.

2. the finding of improved perfusion is consistent with the finding that NEC is improved. Missing is evidence that the improved perfusion is the reason for the improvement in NEC, and not the result of it.

3. a large, multicenter trial published in the nejm in which 1403 patients were studied for the role of remote preconditioning for heart surgery showed zero benefit for patients. this raises doubts regarding the rationale, and the significance, of the current study.

4. various mechanisms are proposed to explain the benefit of RIC on page 5 – at least one of these could be tested in the current model, strengthening the current work.

5. Figure 1 – in the study design, the early RIC actually occurs after the first time point in the late ric group. this makes interpretation difficult as to what is early vs. late.

6. Figure 1 – what happens to RIC when added to BF pups alone? Any alterations in histology? IL-6?

7. Figure 1i – the survival data is potentially profound, but it looks as though the differences are reflective of a small number of pups – please address how many pups.

8. Figure 2 – the middle panel appears to be cut in a very different plane than all other panels; the perfusion index doesn't seem reflective.

9. Figure 3 – these images are beautiful but do not add to the underlying mechanism. the mech by which RIC improves perfusion is not tested.

Reviewer #2:

Remarks to the Author:

This study examined the effect of RPC on vascular function in an induced murine model of NEC.

Conceptually there is modest novelty since RIC has previously been applied to protection of gastric ischemia and improved mucosal blood flow. Here the role of H₂S is examined. Several major concerns limit enthusiasm for this paper.

Major concerns

1. Vascular function and ultimately tissue perfusion are the key outcome measures in this study. However this is measured by anatomic assessment of vessel density and volumetric flow through

A1 vessels. Anatomic assessment of vessel size does not necessarily translate into changes in nutritive perfusion and is not a quantitative measure of flow. Similarly, changes in feed artery flow could be directed to tissue perfusion or be diverted through AV shunts, without improving perfusion. Use of microspheres or even laser Doppler assessment of flow would better address these issues.

2. Arteriole diameter and velocity and flow volume are measured at baseline. To adequately assess whether perfusion is sufficient, some stress is needed (e.g. baseline and max flow following adenosine infusion). Loss of flow reserve would better indicate an ischemic environment. Alternatively, measuring venous oxygen saturation or lactate production would help establish presence of ischemia.

3. The bulk of the presented data are observational. Only figure 4 begins to address mechanism but the data are not as robust as expected. H2S synthase inhibitors make NEC grade worse but no data are provided to suggest this is related to changes in flow. Administration of H2S reduces NEC. However there are key missing experiments. First, can H2S rescue the effect of H2S synthesis inhibitors? Second, what is happening to H2S levels in the tissue during these interventions? Third, does scavenging of H2S (e.g. vitamin B12) recapitulate the effects of H2S synthase inhibitors? The authors could also use genetic models devoid of H2S inhibitors to more selectively test their role in NEC. Finally there is no control for the potent vasodilating effects of H2S. Use of NO, adenosine, or nitroprusside in figure 4e would help establish specificity for H2S.

4. It is presumed by the authors that the mechanism of early and late RPC is the same since they are equally effective. This logic is flawed and only assessment of early RPC mechanism can be claimed.

5. An important missing control is to determine the effect of RIC and H2S inhibitors in control breast fed animals.

Other comments

- Much of the results under "mechanism" are really just further characterization of the NEC process. For example, changes in vascular density, height, leukocyte adhesion, arteriolar diameter and flow are all phenotypical changes that occur with NEC and may be improved with IPC but no data are presented to show they are mechanistic rather than simply coincident with NEC.

Reviewer #3:

Remarks to the Author:

This report studies RIC in a murine model of neonatal enterocolitis. RIC is tested "early" (pre RIC days 5 and 7) and "late" (Days 6 and 8; pre and post RIC) RIC is remarkably effective and the mechanism involves improved microperfusion. Improved perfusion with RIC has been shown previously in both brain and coronary circulations as the authors reference. The methods are elegant. This work is highly translatable as the authors add to the evidence that RIC is safe and effective

Some questions

- 1.) Was RIC applied on one or both hindlimbs? Not clear in lines 305-8
- 2.) How was this regimen of RIC chosen? They were certainly effective but how were the regimens arrived at?
- 3.) The authors studied the H2S system. The NO and NOS system has also been implicated in the mechanism of action of RIC and are important in perfusion. Were NOS 3 inhibitors tested or NOS3 KO mice? The authors might comment on this

Remote ischemic conditioning counteracts the intestinal damage of necrotizing enterocolitis by improving intestinal microcirculation

Response to Reviewer Comments

We would like to thank the editor and the reviewers for their careful review of our manuscript and for the thoughtful, and constructive suggestions. We have performed several additional experiments that further reinforce our initial findings. Our responses to reviewers' comments are in blue in the text below. The revisions that address the reviewers' comments are highlighted in yellow in the manuscript.

Reviewers' comments:

Reviewer #1 (Remarks to the Author):

The goals of this paper are to determine the effects of remote ischemic conditioning on experimental NEC, and to determine its mechanism of action.

Pups underwent early or late RIC and noted improved survival and reduced inflammation.

This is an interesting study. I have the following concerns.

1. ischemic preconditioning has been shown to reduce inflammatory responses in many animal models of sepsis and inflammation (see for ex PMID: 30809283, PMID 28437377, PMID 26436208, PMID 24904237, PMID 25037959) among others; these papers offer significant mechanistic insights that could be tested in the current model.

We appreciate this comment from the reviewer. In the present study, we have identified that the mechanism of action of RIC is the restoration of intestinal perfusion through enhanced vasodilation. We have validated this by using inhibitors of the vasodilatory gasotransmitter, hydrogen sulfide (H₂S), H₂S scavengers, and inhibitors of Nitric Oxide (NO), the downstream effector of hydrogen sulfide (Fig. 5, 6). In addition, we investigated the role of NO in the mechanism of action of RIC by studying the effect of vasodilation in eNOS knockout mice (Supplementary Fig. 5). Our findings demonstrate a critical role for the vasodilatory action of H₂S and NO in the RIC-mediated preservation of intestinal perfusion, leading to a reduction in intestinal injury and inflammation and enhancing survival during experimental NEC (lines 199-268).

2. The finding of improved perfusion is consistent with the finding that NEC is improved. Missing is evidence that the improved perfusion is the reason for the improvement in NEC, and not the result of it.

Thank you for this comment. We found that the mechanism of action of RIC is restoration of intestinal perfusion during experimental NEC, which is mediated through

vasodilation and leads to an improved outcome of NEC. In order to validate this, we have targeted the vasodilatory action of hydrogen sulfide (H₂S), which has been demonstrated previously to have a critical role in improving intestinal perfusion during experimental NEC¹. We have used inhibitors of endogenous H₂S-synthesizing enzymes, H₂S scavengers, as well as inhibitors of NO. Our findings demonstrate that inhibition of vasodilation renders RIC ineffective in improving intestinal wall perfusion (Fig. 5a-b) and intestinal microcirculation (Fig. 5d-e, g-h). As a result of this, NEC-induced intestinal injury and inflammation are also not improved (Fig. 6), ultimately leading to the same mortality observed in NEC alone (Fig. 7). These findings provide evidence that the RIC-mediated improvement in intestinal perfusion is not secondary to improved intestinal morphology and is required to counteract the effects of NEC (lines 242-268).

3. A large, multicenter trial published in the *nejm* in which 1403 patients were studied for the role of remote preconditioning for heart surgery showed zero benefit for patients. this raises doubts regarding the rationale, and the significance, of the current study.

The reviewer is right in pointing out that the effect of preconditioning in improving outcome after heart surgery is controversial.

The rationale of our study is based on the fact that necrotizing enterocolitis (NEC) has an intestinal ischemic component in its pathophysiology. RIC has been applied in many different settings in both humans and animals, in which ischemic injury was involved. Clinical trials have been performed in adults²⁻⁴ and in children⁵⁻⁹ indicating benefits from RIC in various organs including heart, lung, and kidney. In addition, a systematic review and meta-analysis evaluating randomized trials, found that compared with controls, RIC significantly reduced the recurrence of stroke or transient ischemic attacks¹⁰. However, the advantage of RIC remains controversial as two large trials have shown no improvement in relation to cardiac surgery^{11,12}. However, only two trials in adults have focused on the effects of RIC on the intestine. One trial indicated benefit after abdominal aortic aneurism repair when the intestinal ischemia/reperfusion injury is expected¹³ and the other trial indicated no intestinal changes after cardiopulmonary bypass when the intestinal injury is rare, moderate and transient¹⁴. To our knowledge, the potential benefits of RIC in preterm infants and particularly in those with NEC have not been investigated. Our results indicate that RIC is remarkably effective in blunting intestinal ischemic damage in neonatal pups, justifying further investigation of its effectiveness in human preterm infants.

To further clarify our rationale, we added a section in the Introduction, summarizing the information above (lines 96-106).

4. Various mechanisms are proposed to explain the benefit of RIC on page 5 – at least one of these could be tested in the current model, strengthening the current work.

Thank you for this comment. Previous experimental studies in the heart and the brain have demonstrated that remote ischemic conditioning targets the microcirculation in distant ischemic organs¹⁵⁻¹⁸. According to these findings, we have demonstrated that RIC

improves intestinal perfusion and microcirculation via the action of endogenous vasodilatory gasotransmitters including hydrogen sulfide and nitric oxide. To further explore the mechanism of action of RIC (Fig. 5-7), we performed additional experiments using inhibitors of hydrogen sulfide and nitric oxide as well as eNOS knockout mice. We demonstrated that inhibition of vasodilation eliminates the protective effect of RIC characterized by normalization of the intestinal perfusion (Fig. 5), elimination of the intestinal epithelial damage and inflammation (Fig. 6), and ultimately, improvement in survival (Fig. 7) (lines 199-268).

5. Fig. 1 – in the study design, the early RIC actually occurs after the first time point in the late ric group. this makes interpretation difficult as to what is early vs. late.

We appreciate this comment from the reviewer. Previous studies have suggested that remote ischemic conditioning activates two distinct time frames of protection against ischemia reperfusion (IR) injury in the brain and heart. The initial window of protection occurs immediately after the RIC stimulus and lasts for 2 hours, whereas the second window of protection occurs 12-24 hours after the RIC stimulus and lasts 48-72 hours^{19,20}. Moreover, it has been reported that compared with a single episode of remote ischemic conditioning, repeated episodes were more protective in reducing inflammation in the ischemic myocardium²¹. Based on such findings, we chose distinct time points of RIC, 48 hours apart, through the course of our disease model.

To provide further clarification on this important point, we have modified the nomenclature of the experimental groups:

- i. α RIC: RIC was given just before induction of NEC on P5 and repeated 48 hours later on P7 when the intestinal damage was not yet present or it was minimal. This simulates human Stage IIA NEC which is characterized by mild disease according to modified Bell's classification²²⁻²⁴.
- ii. β RIC: RIC was given during NEC development (P6 and P8) when changes in intestinal damage start to be detectable. This simulates Stage IIB NEC which is characterized by moderate disease²²⁻²⁴.

RIC was extremely effective during both α RIC and β RIC in experimental NEC.

These findings are very important for the translational application of RIC in human infants with NEC as RIC can be applied to those at the initial stages of the disease. On the basis of these novel experimental observations, we plan a multicentre randomised controlled trial to prove efficacy of RIC in Stage IIA or IIB NEC. The publication of the present manuscript is of utmost importance in the design and the execution of this trial.

To clarify the experimental study design, we have added a section in the manuscript, summarizing the information above. Please see discussion (lines 305-319) and methods (lines 411-418).

6. Fig. 1 – what happens to RIC when added to BF pups alone? Any alterations in histology? IL-6?

We have performed the suggested experiments and demonstrated no alterations in intestinal histology and IL-6 when RIC was given to breastfed pups (Supplementary Fig. 1) (lines 131-133).

7. Fig. 1i – the survival data is potentially profound, but it looks as though the differences are reflective of a small number of pups – please address how many pups.

We appreciate this critical comment from the reviewer. In our initial experimental design, pups were sacrificed at P9. We increased the sample size to include at least 20 pups per group and demonstrated that survival up to P9 was enhanced by either α RIC or β RIC (Fig. 7) in wild type. However, this beneficial effect was not seen in wild type pups given inhibitors of H₂S (Fig. 7), or in eNOS knockout pups (Supplementary Fig. 5d).

In addition, to further characterize the outcome after RIC, we performed additional experiments aimed at quantifying the survival rate after P9. Pups in the various experimental groups continued to receive gavage feeding and were observed by an investigator blinded to treatment allocation until death occurred. Compared to NEC alone, the survival was extended by either α RIC or β RIC (Fig. 7). Conversely, inhibitors of H₂S resulted in a similar mortality rate to NEC alone. These important findings reinforce our observations (Fig. 5 and 6) on the mechanism of action of RIC being dependent on the H₂S pathway. (Fig. 7) (lines 256-266).

8. Fig. 2 – the middle panel appears to be cut in a very different plane than all other panels; the perfusion index doesn't seem reflective.

We have corrected the middle panel to match the others and added indicators in Fig. 2. In addition, we clarified that the perfusion index refers to the ratio of the intra-villi arteriole to the whole villi (lines 494-499) indicating significant differences between NEC alone and NEC with RIC.

9. Fig. 3 – these images are beautiful but do not add to the underlying mechanism. the mech by which RIC improves perfusion is not tested.

In the revised manuscript, we have provided further evidence to explain the underlying mechanism of action of RIC by performing various new experiments. Our findings suggest that RIC improves the outcome of NEC, enhances survival, and restores the NEC-induced derangements in intestinal perfusion via a vasodilatory-dependent mechanism. We have demonstrated this by assessing changes in the RIC-mediated restoration of intestinal perfusion upon treatment with inhibitors of vasodilatory mediators such as hydrogen sulfide and nitric oxide. Our findings suggest that inhibition of endogenous synthesis of hydrogen sulfide through administration of H₂S-synthesizing enzyme inhibitors, and scavenging of H₂S, both considerably abolish the protection

conveyed by RIC during experimental NEC, leading to impaired perfusion, increased intestinal injury and inflammation, and poor survival (Fig. 5-7). Our findings also suggest that inhibition of NO-mediated vasodilation abolishes the RIC-mediated improvements in intestinal wall perfusion and increases the intestinal injury. Taken together, the mechanism by which RIC improves intestinal perfusion during experimental NEC is the collective vasodilatory action of gasotransmitters such as H₂S and NO.

We have added the above explanation in the manuscript (lines 199-268).

Reviewer #2 (Remarks to the Author):

This study examined the effect of RPC on vascular function in an induced murine model of NEC. Conceptually there is modest novelty since RIC has previously been applied to protection of gastric ischemia and improved mucosal blood flow. Here the role of H₂S is examined. Several major concerns limit enthusiasm for this paper.

Major concerns

1. Vascular function and ultimately tissue perfusion are the key outcome measures in this study. However this is measured by anatomic assessment of vessel density and volumetric flow through A1 vessels. Anatomic assessment of vessel size does not necessarily translate into changes in nutritive perfusion and is not a quantitative measure of flow. Similarly, changes in feed artery flow could be directed to tissue perfusion or be diverted through AV shunts, without improving perfusion. Use of microspheres or even laser Doppler assessment of flow would better address these issues.

We thank the reviewer for the comments. To assess changes in arterial flow, we measured flow velocity in the entire intestinal wall using Doppler Ultrasound^{25,26}. Intestinal wall perfusion was calculated as average flow velocity (mm/s) of multiple abdominal regions (Fig. 4a). In agreement with intestinal damage (Fig. 1a) and decreased microcirculation (Fig. 3), intestinal wall flow velocity (Fig. 4) was significantly reduced in NEC pups, compared to breastfed controls. In contrast, both α RIC and β RIC increased flow velocity in the intestinal wall (Fig. 4) demonstrating improved intestinal perfusion (Supplementary movies 9-12). Conditioning with α RIC or β RIC in breastfed control pups did not alter intestinal wall flow velocity (Supplementary Fig. 1f). We added these results in the manuscript (lines 187-195).

These findings are in agreement with the changes observed in intestinal microcirculation using two photon laser scanning microscopy (TPLSM) (lines 187-195).

2. Arteriole diameter and velocity and flow volume are measured at baseline. To adequately assess whether perfusion is sufficient, some stress is needed (e.g. baseline and max flow following adenosine infusion). Loss of flow reserve would better indicate an ischemic environment. Alternatively, measuring venous oxygen saturation or lactate production would help establish presence of ischemia.

As suggested, we have performed additional experiments using two photon laser scanning microscopy (TPLSM) to assess submucosal arteriole diameter, velocity, and flow volume at baseline in P5 and P9 pups in response to formula feeding as a stress factor (Fig. 3b-d). Our findings indicate that single formula feeding at P5 caused no significant change in arteriole velocity, diameter, and flow volume. On the contrary, later in the neonatal period (P9), formula feeding resulted in significant increase over baseline in arteriole velocity, diameter, and flow volume. Taken together, these findings suggest that the immature submucosal arterioles of the intestine respond poorly to feeding in early neonatal mice (P5), which could contribute to feeding-induced intestinal hypoxia and development of NEC. Excitingly, RIC is able to counteract the poor response to feeding in early neonatal mice by improving intestinal microcirculation, leading to protection against NEC development and ultimately enhanced survival. Please see Fig. 3b-d for above results (lines 163-171).

We agree on the importance of assessing intestinal ischemia. Pimonidazole is a sensitive marker of intestinal ischemia and most importantly it allows the localization of ischemia²⁷. In accordance with previous work from our group²⁷, we found that NEC is associated with ischemia at the tip of the villi (Fig. 2). We performed additional experiments and discovered lack of vascular flow at the tip of the villi explaining the ischemia occurring in this area. RIC during experimental NEC improves the flow to the top of the villi, thus avoiding ischemia (Fig. 2).

3. The bulk of the presented data are observational. Only Fig. 4 begins to address mechanism but the data are not as robust as expected. H₂S synthase inhibitors make NEC grade worse but no data are provided to suggest this is related to changes in flow. Administration of H₂S reduces NEC. However there are key missing experiments. First, can H₂S rescue the effect of H₂S synthesis inhibitors? Second, what is happening to H₂S levels in the tissue during these interventions? Third, does scavenging of H₂S (e.g. vitamin B12) recapitulate the effects of H₂S synthesis inhibitors? The authors could also use genetic models devoid of H₂S inhibitors to more selectively test their role in NEC. Finally, there is no control for the potent vasodilating effects of H₂S. Use of NO, adenosine, or nitroprusside in Fig. 4e would help establish specificity for H₂S.

We appreciate the above comments from the reviewer, and we have performed various additional experiments to address these questions.

- (1) We investigated whether administration of NaHS, an exogenous H₂S donor, can rescue the effect of H₂S synthesis inhibitors. In Fig. 5 and 6, following treatment with H₂S synthesis inhibitors, NaHS did not improve intestinal morphology and did not reduce intestinal inflammation. Likewise, NEC-induced derangements in intestinal wall perfusion and impaired velocity, diameter, and flow volume of submucosal arterioles were not rescued (lines 199-268).
- (2) We have performed additional experiments to validate changes in expression of H₂S in the ileum during these interventions. Study of H₂S under *in vivo* conditions

- is challenging due to the short half-life of H₂S²⁸. In addition, following our consultation with experts in the field, we learned that direct measurement of H₂S levels in tissue is not an ideal approach. Hence, we performed immunofluorescence staining for cystathionine-β-synthase (CBS), one of the key endogenous H₂S-synthesizing enzymes in the ileum. Please see Supplementary Fig. 2 and the corresponding explanation of this data in the manuscript (lines 201-204).
- (3) As suggested, we have performed additional experiments to study whether scavenging of H₂S via the vitamin B12 analog, recapitulates the effects of H₂S synthesis inhibitors. Our findings suggest that treatment with the vitamin B12 analog produced similar results to H₂S synthesis inhibitors. These findings are reported in the results (lines 199-268) and in Fig. 5 and 6.
 - (4) As H₂S synthesis is regulated via three enzyme pathways, the use of selective genetic mutant mice to study the overall role of H₂S in RIC and NEC can be complicated. For this reason, we have been using H₂S synthesis inhibitors and H₂S scavengers. In addition, H₂S has been reported to improve mesenteric perfusion and intestinal injury in experimental NEC via an eNOS-dependent mechanism¹, suggesting that NO-mediated regulation of intestinal microcirculation occurs downstream of H₂S. We performed additional experiments using eNOS knockout mice and demonstrated that RIC in these mice had no beneficial effect on the NEC-induced intestinal injury (morphology and inflammation), and eventually did not improve survival. These data are presented in results (lines 225-238, 252-255), and in Supplementary Fig. 5.
 - (5) To further investigate the role of NO in mediating the effects of RIC on the perfusion of the entire intestinal wall and more deeply of the submucosal microcirculation, we used the NO synthase inhibitor, L-NAME. As indicated in Fig. 5 and 6, treatment with L-NAME abolished the RIC-mediated improvement in the perfusion of both the intestinal wall and the submucosal microcirculation (lines 225-238, 252-255).

4. It is presumed by the authors that the mechanism of early and late RPC is the same since they are equally effective. This logic is flawed and only assessment of early RPC mechanism can be claimed.

We have modified the nomenclature of early and late RIC to α RIC (P5 and P7) when the intestinal damage was not yet present or it was minimal; and β RIC (P6 and P8) when changes in intestinal damage start to be detectable. We performed additional experiments to evaluate the effects of α RIC as well as β RIC. Our results demonstrated that α RIC and β RIC were equally effective in NEC and had the same mechanism of action (Fig. 5-7).

5. An important missing control is to determine the effect of RIC and H₂S inhibitors in control breast fed animals.

We have performed additional experiments and studied the effect of RIC and H₂S inhibitors in breastfed controls. We have not observed any change in intestinal morphology, inflammation, and intestinal perfusion by RIC on breastfed control pups. These findings are shown in Supplementary Fig 1.

Other comments

- Much of the results under “mechanism” are really just further characterization of the NEC process. For example, changes in vascular density, height, leukocyte adhesion, arteriolar diameter and flow are all phenotypical changes that occur with NEC and may be improved with IPC but no data are presented to show they are mechanistic rather than simply coincident with NEC.

Thank you very much for your comments. We have found that the mechanism of action of RIC is restoration of intestinal perfusion during experimental NEC, which is mediated through vasodilation and leads to improved NEC outcome. To validate this, we have targeted the vasodilatory action of H₂S, which has been demonstrated by previous authors to have a critical role in improving intestinal perfusion during experimental NEC¹. We have used inhibitors of endogenous H₂S synthesizing enzymes, H₂S scavengers, as well as inhibitors of nitric oxide, a downstream effector of H₂S. Our findings demonstrate that inhibition of vasodilation renders RIC ineffective in improving intestinal wall perfusion (Fig. 5a-b) and intestinal microcirculation (Fig. 5d-e, g-h). As a result of this, NEC-induced intestinal injury and inflammation are also not improved (Fig. 6), ultimately leading to reduced survival (Fig. 7). These findings provide evidence that the RIC-mediated improved perfusion is required for a consequent improvement in NEC. This is addressed in the manuscript (lines 199-268).

Reviewer #3 (Remarks to the Author):

This report studies RIC in a murine model of neonatal enterocolitis. RIC is tested "early" (pre RIC days 5 and 7) and "late" (Days 6 and 8; per and post RIC) RIC is remarkably effective and the mechanism involves improved microperfusion. Improved perfusion with RIC has been shown previously in both brain and coronary circulations as the authors reference. The methods are elegant. This work is highly translatable as the authors add to the evidence that RIC is safe and effective

Some questions

1.) Was RIC applied on one or both hindlimbs? Not clear in lines 305-8

We appreciate the comment from the reviewer. The RIC stimulus was always given to one hind limb and the same hind limb was used for all pups throughout the various experiments. We have clarified this in the manuscript (lines 408-409)

2.) How was this regimen of RIC chosen? They were certainly effective but how were the regimens arrived at?

The phenomenon of ischemic pre-conditioning was first described in the canine heart wherein four 5 min circumflex coronary occlusions, each separated by 5 min of reperfusion, dramatically reduced myocardial infarction size²⁹. Przyklenk et al. also showed that brief myocardial ischemia by four cycles of 5 min coronary artery occlusion protected local and remote myocardium from sustained 1 h cardiac ischemia reperfusion injury³⁰. Experimental and clinical evidence also suggests that RIC activates at least two distinct time frames of protection against ischemia reperfusion injury of the brain and heart^{19,20}. The first window of protection occurs immediately after the RIC stimulus and lasts for 2 hours, and involves changes in ion channel permeability, protein phosphorylation, and release of several signaling mediators¹⁹. The second window of protection follows 12-24 hours after the RIC stimulus and lasts 48-72 hours, involving modulation of inflammatory response, improved endothelial function, and activation of gene expression²⁰. Moreover, previous authors have reported that compared with a single episode of remote ischemic conditioning, repeated episodes were more protective in reducing inflammation in the ischemic myocardium²¹. Hence, the previously used protocol of four cycles of 5 min occlusion followed by 5 min reperfusion was chosen for RIC in our model (lines 305-319).

We selected two time periods of RIC (α RIC at P5 and P7, and β RIC at P6 and P8) which simulate two initial stages of NEC, Stage IIA and Stage IIB respectively^{23,24}. The efficacy of RIC in both time periods is important for the translational application of this experimental observation. We added an explanation in the manuscript (lines 305-319).

3.) The authors studied the H₂S system. The NO and NOS system has also been implicated in the mechanism of action of RIC and are important in perfusion. Were NOS 3 inhibitors tested or NOS3 KO mice? The authors might comment on this

Previous authors have reported that H₂S improves mesenteric perfusion and intestinal injury in experimental NEC via an eNOS-dependent mechanism¹, suggesting that NO-mediated regulation of intestinal microcirculation occurs downstream of H₂S.

To further illustrate the importance of NO in RIC-mediated vasodilation during experimental NEC, we performed additional experiments using eNOS knockout mice. We demonstrated that RIC in these mice had no beneficial effect on NEC intestinal injury (morphology and inflammation) and did not improve survival. The baseline perfusion of various organs, including the intestine, was tenuous due to lack of eNOS and the measurement of intestinal wall perfusion during experimental NEC and RIC was not reliable. To overcome this difficulty, we explored the role of NO synthase inhibitor L-NAME on the effects RIC in experimental NEC. Our findings demonstrated that the RIC-mediated improvement in the perfusion of the whole intestinal wall and of the submucosal layer was abolished upon treatment with L-NAME. Consistent with eNOS knock out mice, inhibition of NO by L-NAME eliminated the RIC beneficial effects of reducing NEC-related intestinal injury and inflammation. The findings from these

additional experiments can be found in the results (lines 225-238, 252-255), and in Fig. 5-7.

References

- 1 Drucker, N. A., Jensen, A. R., Te Winkel, J. P. & Markel, T. A. Hydrogen Sulfide Donor GYY4137 Acts Through Endothelial Nitric Oxide to Protect Intestine in Murine Models of Necrotizing Enterocolitis and Intestinal Ischemia. *The Journal of surgical research* **234**, 294-302, doi:10.1016/j.jss.2018.08.048 (2019).
- 2 Eitel, I. *et al.* Cardioprotection by combined intrahospital remote ischaemic preconditioning and postconditioning in ST-elevation myocardial infarction: the randomized LIPSIA CONDITIONING trial. *European heart journal* **36**, 3049-3057, doi:10.1093/eurheartj/ehv463 (2015).
- 3 Botker, H. E. *et al.* Remote ischaemic conditioning before hospital admission, as a complement to angioplasty, and effect on myocardial salvage in patients with acute myocardial infarction: a randomised trial. *Lancet* **375**, 727-734, doi:10.1016/S0140-6736(09)62001-8 (2010).
- 4 Gaspar, A. *et al.* Randomized controlled trial of remote ischaemic conditioning in ST-elevation myocardial infarction as adjuvant to primary angioplasty (RIC-STEMI). *Basic research in cardiology* **113**, 14, doi:10.1007/s00395-018-0672-3 (2018).
- 5 Kang, Z. *et al.* Remote ischemic preconditioning upregulates microRNA-21 to protect the kidney in children with congenital heart disease undergoing cardiopulmonary bypass. *Pediatric nephrology* **33**, 911-919, doi:10.1007/s00467-017-3851-9 (2018).
- 6 Zhong, H. *et al.* Cardioprotective effect of remote ischemic postconditioning on children undergoing cardiac surgery: a randomized controlled trial. *Paediatric anaesthesia* **23**, 726-733, doi:10.1111/pan.12181 (2013).
- 7 Luo, W., Zhu, M., Huang, R. & Zhang, Y. A comparison of cardiac post-conditioning and remote pre-conditioning in paediatric cardiac surgery. *Cardiology in the young* **21**, 266-270, doi:10.1017/S1047951110001915 (2011).
- 8 Zhou, W. *et al.* Limb ischemic preconditioning reduces heart and lung injury after an open heart operation in infants. *Pediatric cardiology* **31**, 22-29, doi:10.1007/s00246-009-9536-9 (2010).
- 9 Cheung, M. M. *et al.* Randomized controlled trial of the effects of remote ischemic preconditioning on children undergoing cardiac surgery: first clinical application in humans. *Journal of the American College of Cardiology* **47**, 2277-2282, doi:10.1016/j.jacc.2006.01.066 (2006).
- 10 Zhao, J. J. *et al.* Remote Ischemic Postconditioning for Ischemic Stroke: A Systematic Review and Meta-Analysis of Randomized Controlled Trials. *Chinese medical journal* **131**, 956-965, doi:10.4103/0366-6999.229892 (2018).
- 11 Meybohm, P. *et al.* A Multicenter Trial of Remote Ischemic Preconditioning for Heart Surgery. *N Engl J Med* **373**, 1397-1407, doi:10.1056/NEJMoa1413579 (2015).
- 12 Hausenloy, D. J. *et al.* Remote Ischemic Preconditioning and Outcomes of Cardiac Surgery. *N Engl J Med* **373**, 1408-1417, doi:10.1056/NEJMoa1413534 (2015).
- 13 Li, C. *et al.* Limb remote ischemic preconditioning for intestinal and pulmonary protection during elective open infrarenal abdominal aortic aneurysm repair: a

- randomized controlled trial. *Anesthesiology* **118**, 842-852, doi:10.1097/ALN.0b013e3182850da5 (2013).
- 14 Struck, R. *et al.* Effect of Remote Ischemic Preconditioning on Intestinal Ischemia-Reperfusion Injury in Adults Undergoing On-Pump CABG Surgery: A Randomized Controlled Pilot Trial. *Journal of cardiothoracic and vascular anesthesia* **32**, 1243-1247, doi:10.1053/j.jvca.2017.07.027 (2018).
- 15 Kitagawa, K., Saitoh, M., Ishizuka, K. & Shimizu, S. Remote Limb Ischemic Conditioning during Cerebral Ischemia Reduces Infarct Size through Enhanced Collateral Circulation in Murine Focal Cerebral Ischemia. *Journal of stroke and cerebrovascular diseases : the official journal of National Stroke Association* **27**, 831-838, doi:10.1016/j.jstrokecerebrovasdis.2017.09.068 (2018).
- 16 Ren, C. *et al.* Limb Ischemic Conditioning Improved Cognitive Deficits via eNOS-Dependent Augmentation of Angiogenesis after Chronic Cerebral Hypoperfusion in Rats. *Aging and disease* **9**, 869-879, doi:10.14336/AD.2017.1106 (2018).
- 17 Zheng, Y., Lu, X., Li, J., Zhang, Q. & Reinhardt, J. D. Impact of remote physiological ischemic training on vascular endothelial growth factor, endothelial progenitor cells and coronary angiogenesis after myocardial ischemia. *International journal of cardiology* **177**, 894-901, doi:10.1016/j.ijcard.2014.10.034 (2014).
- 18 Kono, Y. *et al.* Remote ischemic conditioning improves coronary microcirculation in healthy subjects and patients with heart failure. *Drug design, development and therapy* **8**, 1175-1181, doi:10.2147/DDDT.S68715 (2014).
- 19 Ren, C., Gao, X., Steinberg, G. K. & Zhao, H. Limb remote-preconditioning protects against focal ischemia in rats and contradicts the dogma of therapeutic time windows for preconditioning. *Neuroscience* **151**, 1099-1103, doi:10.1016/j.neuroscience.2007.11.056 (2008).
- 20 Kuzuya, T. *et al.* Delayed effects of sublethal ischemia on the acquisition of tolerance to ischemia. *Circulation research* **72**, 1293-1299, doi:10.1161/01.res.72.6.1293 (1993).
- 21 Jiang, Q. *et al.* Systemic redistribution of the intramyocardially injected mesenchymal stem cells by repeated remote ischaemic post-conditioning. *Journal of cellular and molecular medicine* **22**, 417-428, doi:10.1111/jcmm.13331 (2018).
- 22 Lee, J. S. & Polin, R. A. Treatment and prevention of necrotizing enterocolitis. *Seminars in neonatology : SN* **8**, 449-459, doi:10.1016/S1084-2756(03)00123-4 (2003).
- 23 Walsh, M. C. & Kliegman, R. M. Necrotizing enterocolitis: treatment based on staging criteria. *Pediatric clinics of North America* **33**, 179-201, doi:10.1016/s0031-3955(16)34975-6 (1986).
- 24 Bell, M. J. *et al.* Neonatal necrotizing enterocolitis. Therapeutic decisions based upon clinical staging. *Annals of surgery* **187**, 1-7, doi:10.1097/00000658-197801000-00001 (1978).
- 25 Choi, Y. H. *et al.* Doppler sonographic findings in an experimental rabbit model of necrotizing enterocolitis. *J Ultrasound Med* **29**, 379-386, doi:10.7863/jum.2010.29.3.379 (2010).

- 26 Faingold, R. *et al.* Necrotizing enterocolitis: assessment of bowel viability with color doppler US. *Radiology* **235**, 587-594, doi:10.1148/radiol.2352031718 (2005).
- 27 Chen, Y. *et al.* Formula feeding and systemic hypoxia synergistically induce intestinal hypoxia in experimental necrotizing enterocolitis. *Pediatric surgery international* **32**, 1115-1119, doi:10.1007/s00383-016-3997-8 (2016).
- 28 Polhemus, D. J. & Lefer, D. J. Emergence of hydrogen sulfide as an endogenous gaseous signaling molecule in cardiovascular disease. *Circulation research* **114**, 730-737, doi:10.1161/CIRCRESAHA.114.300505 (2014).
- 29 Murry, C. E., Jennings, R. B. & Reimer, K. A. Preconditioning with ischemia: a delay of lethal cell injury in ischemic myocardium. *Circulation* **74**, 1124-1136, doi:10.1161/01.cir.74.5.1124 (1986).
- 30 Przyklenk, K., Bauer, B., Ovize, M., Kloner, R. A. & Whittaker, P. Regional ischemic 'preconditioning' protects remote virgin myocardium from subsequent sustained coronary occlusion. *Circulation* **87**, 893-899, doi:10.1161/01.cir.87.3.893 (1993).

Reviewers' Comments:

Reviewer #1:

Remarks to the Author:

Re: remote ischemic conditioning counteracts the intestinal damage of necrotizing enterocolitis by improving intestinal microcirculation.

I have carefully reviewed the revised manuscript.

Each of my original concerns remains, and I have in re-review, identified additional concerns.

Major.

1. The new concern is based upon the fact that the study does not actually provide any evidence that the intestinal microcirculation is improved in the presence of ischemic conditioning. While perfusion was enhanced (see below for the lack of proof that this is a cause and not a simple consequence of NEC improvement), the actual microcirculation i.e. the collection of blood vessels within the wall of the bowel were not shown to be altered in a way that resulted in the improvement in NEC. As such, the major premise of the work, and identified in the title even, was not supported by the data provided.

2. I take issue with their statement that the authors have shown that the restoration of intestinal perfusion is the mechanism involved. Moreover, the use of H₂S scavengers – which blunt the protection of ischemic conditioning – does not logically explain an effect due to perfusion, in as much as H₂S has many effects on the cell and the host, that are independent of any role on perfusion. Only by blocking perfusion specifically and losing the effects of preconditioning, can a link be made. It is true that there are effects of NO and H₂S in the current studies, but these may be related to immune effects which secondarily affect perfusion, given the pleiotropic roles of these second messengers.

3. my original review included some 5 prior instances in which ischemic preconditioning reduces inflammation; I remain concerned that the work is a logical extension of the work of others.

4. I remain very concerned by the data in Figure 1 that the early RIC occurs actually after the first time point in the late RIC group, so the data regarding early vs. late is uninterpretable. If the authors now believe that both early and late RIC is effective, given the fact that longer durations of RIC would be expected to have greater effects on H₂S or NO production, I'm even more concerned by the results.

5. I remain concerned by the number of repeats and whether the data in Figures 2-3 and also 5-7 are biological vs. technical vs. experimental repeats.

6. why is all RT-PCR data shown in aggregate as bars and not scatter plots? This is concerning given the high error bars.

Reviewer #2:

Remarks to the Author:

The authors should be commended for providing extensive new data, however some of the data does not address the initially raised concerns regarding mechanism of RIC effect, and thus overall enthusiasm for the manuscript is only modestly improved.

Top of page 15: The authors treat with H₂S synthesis inhibitors and show lack of protection by RIC against NEC. They conclude that the RIC effect is due to changes in intestinal perfusion. This is not

an acceptable conclusion from these data. It is highly possible that H₂S is acting in some other way (than dilation) to mediate its beneficial effect since H₂S has a host of other biochemical and physiological effects. Flow changes may be secondary, not causative. The problem is that the experiments necessary to prove that it changes in flow are responsible, are difficult to do. One would need to add H₂S or nitric oxide and include a vasoconstrictor to prevent changes in flow to show that the beneficial effect of these agents on NEC was abrogated. In addition use of a nonspecific vasodilator such as Papaverine should be used to show that improved perfusion is sufficient to inhibit the effects of NEC. Short of such data (that dissociate changes in flow and other signaling actions of H₂S and NO) the authors cannot conclude that changes in flow are mechanistically responsible. The problem with using inhibitors of H₂S formation or of NO production (LNAME or eNOS KO) is that you end up blocking both the vasomotor effects and the many other biochemical signaling effects of these molecules, thus you cannot distinguish which is responsible for improvement in NEC via RPC.

Response to Reviewer's Comments

Remote ischemic conditioning counteracts the intestinal damage of necrotizing enterocolitis by improving intestinal microcirculation

We would like to thank the editor and the reviewers for their careful review of our revised manuscript and for the thoughtful and constructive comments.

We have performed various additional experiments in both human and mice that further reinforce our initial findings. Before answering point-by-point the comments raised by the reviewers, we would like to briefly summarize (highlighted in blue) the substantial additions and changes made in the resubmitted manuscript. These stem from comments received by the reviewers as well as our motivation to make this discovery clearer and applicable to humans.

Please note that we have highlighted (in yellow) the additional experiments and modification in our resubmitted manuscript.

Summary of Added Experiments in Resubmission

1. Studies in human neonates with NEC and control neonates without NEC: These studies indicate intestinal microcirculatory deficiency in NEC.
2. Investigation of the role of RIC in severe NEC: This required a novel set of *in vivo* and *in vitro* experiments. The results obtained demonstrate that RIC is beneficial in the initial stages of the disease but not when severe damage has already occurred. Clarifying the importance of the timing of RIC further supports its mechanism of action while providing important data for translation of this novel therapy into human neonates with NEC.
3. Further exploration of the effect of RIC on microcirculation by administering nonspecific vasodilator and vasoconstrictor agents. The results obtained support our previous results obtained using *eNOS* knockout mice as well as chemical inhibitors and donors of NO and H₂S.

Responses

Our responses to the reviewers' comments are in blue in this letter.

Reviewer #1 (Remarks to the Author):

Re: remote ischemic conditioning counteracts the intestinal damage of necrotizing enterocolitis by improving intestinal microcirculation.

I have carefully reviewed the revised manuscript.

Each of my original concerns remains, and I have in re-review, identified additional concerns.

Major.

1. The new concern is based upon the fact that the study does not actually provide any evidence that the intestinal microcirculation is improved in the presence of ischemic conditioning. While perfusion was enhanced (see below for the lack of proof that this is a cause and not a simple consequence of NEC improvement), the actual microcirculation i.e. the collection of blood vessels within the wall of the bowel were not shown to be altered in a way that resulted in the improvement in NEC. As such, the major premise of the work, and identified in the title even, was not supported by the data provided.

We appreciate this concern from the reviewer. Considering the size of our experimental pups at postnatal days 5 to 9, it is not possible to isolate different areas of the intestine. Hence, the experiments necessary to directly detect blood flow and prove alterations in flow are challenging. Nonetheless, the data in our manuscript as listed below provides support that improved intestinal microcirculation due to RIC is responsible for improving the outcome of NEC:

- In Figure 1, we demonstrate that human neonates with NEC have the lowest expression of vascular endothelial marker (cluster of differentiation, CD31) and highest expression of hypoxia marker (Hypoxia-inducible factor 1 α , HIF1 α)¹ in the most affected intestinal area. Ileum farther away from this most affected area regains similar level of CD31 expression and shows reduced HIF1 α expression as compared to non-NEC control neonates. Hence, this data suggests that human NEC is associated with mucosal hypoxia and reduced number of endothelial cells suggestive of compromised intestinal perfusion.
- In Figure 2 and Figure S1b-c, we demonstrate that while RIC in wildtype mice improves the intestinal injury of NEC, reduces inflammation, and enhances survival, RIC is unable to promote the same protective effects in *eNOS* knockout mice. This data suggests that the protection conveyed by RIC in the intestine with NEC-induced injury is dependent on the endothelium and likely relies on endothelium-mediated vasodilation.

- Given that *eNOS* signaling is essential for the RIC-mediated protection against intestinal damage of NEC, we then use Doppler ultrasound to measure intestinal wall perfusion daily during the five days of our NEC induction protocol in mouse pups (Figure 3). Intestinal wall perfusion is calculated as average flow velocity (mm/s) of multiple abdominal regions and is indicative of blood flow within the wall of the abdomen. Using these measurements, we demonstrate the following points:
 - As illustrated in Figure 2a and Figure S1a, there is progressive increase in intestinal injury and inflammation with significant morphological changes and elevated inflammation detectable from P7 (* $p < 0.05$). However, as evident from Figure 3b, intestinal wall flow velocity shows significant reduction in NEC pups from P6 (* $p < 0.001$; statistical analysis not reported in the resubmitted manuscript but can be added upon request). Hence, we detect reduced perfusion in the intestinal wall even before the intestinal epithelium is damaged. This data also demonstrates that reduced perfusion is a contributing factor to the development of NEC, rather than a consequence of NEC; this is because alterations in intestinal perfusion precedes the changes in intestinal morphology and inflammation that are associated with NEC.
 - Using daily measurements of intestinal wall perfusion with Doppler ultrasound (Figure 3), we also demonstrate that in NEC pups, intestinal perfusion remains low from P5 to P9 while the non-NEC breastfed controls show increased perfusion daily. Videos S1-2 depict the derangements in intestinal wall perfusion in the NEC pups, compared to non-NEC breastfed controls. However, administration of RIC to NEC pups significantly enhances perfusion across the intestinal wall, resembling a trend similar to what is seen in the breastfed controls. Please review this data as depicted in Videos S3-4. These findings demonstrate that by manipulating intestinal blood flow with RIC during the course of NEC development, we are able to preserve perfusion and improve the outcome of NEC.
- Videos S5-8 demonstrate our evaluation of intestinal microcirculation in the submucosa using two photon laser scanning microscopy (TPLSM) which allows *in vivo* visualization and quantification of blood flow in real time. We have also quantified blood flow from videos obtained by TPLSM, as shown in Figure 4, to demonstrate that arteriole velocity, diameter, and flow volume in the submucosa are reduced in NEC (Please see Videos S5-6), which is suggestive of impaired perfusion. However, administration of RIC to NEC pups mitigates these derangements and preserves the velocity, diameter, and flow volume of submucosal arterioles (Please see Videos S7-8). Hence, we demonstrate that RIC improves perfusion not just across the intestinal wall as seen with Doppler ultrasound, but also in the intestinal submucosa.
- In Figure 5, we demonstrate that in agreement with preservation of submucosal perfusion by RIC, the integrity of the villi microvasculature is also improved. NEC pups demonstrated increased ischemia at the tip of the villi, evident by increased staining of

the hypoxia marker pimonidazole. Consistently, NEC pups also demonstrated a marked constriction and decrease in the height of arterioles perfusing the villi. Finally, there was increased staining with the necrosis marker, Sytox Green, in the NEC pups, especially at the tip of the villi. These observations are suggestive of impairment in the integrity of intra-villi arterioles, which results in increased hypoxia at the villi tip, and hence necrosis of enterocytes. However, administration of RIC to NEC pups resulted in recovery of the diameter and height of intra-villi arterioles, resulting in reduced hypoxia and necrosis at the villi tip. This data adds to our previous findings and provides further support for the contention that RIC confers protection in the NEC intestine by mitigating the impairments in the microvasculature of the intestine.

Collectively, our data provides evidence that RIC targets the intestinal microcirculation and restores perfusion at the level of the arterioles in the villi, arterioles in the submucosa, as well as perfusion across the entire intestinal wall.

2. I take issue with their statement that the authors have shown that the restoration of intestinal perfusion is the mechanism involved. Moreover, the use of H₂S scavengers – which blunt the protection of ischemic conditioning – does not logically explain an effect due to perfusion, in as much as H₂S has many effects on the cell and the host, that are independent of any role on perfusion. Only by blocking perfusion specifically and losing the effects of preconditioning, can a link be made. It is true that there are effects of NO and H₂S in the current studies, but these may be related to immune effects which secondarily affect perfusion, given the pleiotropic roles of these second messengers.

We appreciate the reviewer's comments regarding the mechanism of action for restoration of perfusion by RIC. We acknowledge the possibility that administration of chemical inhibitors of NO and H₂S blocks not just the effects due to vasoregulation, but possibly other biochemical and physiological effects of these gasotransmitters as well. However, the additional experiments in our newly submitted manuscript address this concern as explained below:

- As mentioned above, we tested the effectiveness of RIC in *eNOS* knockout pups to investigate whether the beneficial effects of RIC for avoiding the intestinal damage of NEC are dependent on the endothelium. In Figure 2 and Figure S1b-c, we demonstrate that unlike in wildtype mice, administration of RIC to *eNOS* knockout pups with NEC fails to improve intestinal morphology, reduce inflammation, or enhance survival. This data provides evidence that the endothelium plays an essential role in the mechanism of action of RIC.
- In Figure S7, we demonstrate that following administration of methoxamine, a general and nonspecific vasoconstrictor², the protective effects of RIC in the NEC intestine are lost. This additional data provides further support for the beneficial effects of RIC due to regulation of blood flow via vasodilation.

- In Figure S7, we also demonstrate that administration of two nonspecific vasodilators, papaverine and captopril, to NEC pups improves intestinal injury, reduces inflammation, and enhances survival; hence providing the same protective effects conferred by RIC. These findings provide evidence that vasodilation is sufficient to improve the outcome of NEC.

3. my original review included some 5 prior instances in which ischemic preconditioning reduces inflammation; I remain concerned that the work is a logical extension of the work of others.

We appreciate the reviewer's concern regarding the effects of ischemic conditioning on inflammation. Data in our newly submitted manuscript provides evidence that changes in intestinal blood flow occur prior to inflammation and injury in the epithelium (Figure in this letter). Hence, derangements of blood flow play a primary role in the development of NEC. Our evidence for this contention is listed below:

- We have previously demonstrated that in the early neonatal period (P5), pups fail to stimulate an increase in intestinal blood flow in response to a single gavage formula feeding³. This inadequate response to feeding stems from the immaturity of the intestinal microvasculature and prevents the intestine from meeting its increased oxygen demand after feeding³. Hence, feeding continues to disturb the balance between oxygen demand and supply in the premature intestine, leading to hypoxia, and the development of NEC³. These previous findings contribute to our understanding of the emergence of the impairments in intestinal microcirculation and hypoxia that are known to be associated with NEC⁴⁻⁸. Collectively, the alterations in intestinal blood flow dynamics which arise due to prematurity and feeding-induced hypoxia are primary events in the development of NEC. Moreover, alterations in intestinal blood flow occur prior to any changes in intestinal morphology or inflammation.

In the current study, in Figure S2a-c, using TPLSM, we demonstrate that administration of RIC to P5 pups offsets the poor microcirculatory response to feeding. Administration of RIC to pups in the early neonatal period enables increased blood flow in the intestinal

microvasculature following feeding, resembling the response seen in mature pups at P9. Quantification of blood flow from videos obtained in real time by TPLSM revealed enhanced velocity, diameter, and flow volume of submucosal arterioles following feeding of P5 pups that received RIC. Hence, we show that by modulating the immature intestinal microcirculation and improving blood flow at an early stage, we can prevent the development of NEC. This data provides further evidence that modulating the blood flow dynamics in the intestine, even before any injury or inflammation has occurred, plays a primary role in improving the outcome of the disease.

- In our resubmitted manuscript, we added various experiments to evaluate whether RIC is still effective if administered in late stages of NEC development, when significant injury and inflammation in the intestine have already been established. As mentioned above in our response to the first comment, morphological changes and increase in inflammation are first detected at P7 during the development of NEC. In Figure 2 of the resubmitted manuscript and the Figure in this letter, we demonstrate that administration of Stage 3 RIC at this point (P7), when NEC-induced injury and inflammation in the intestine are already established, is not able to reverse the injury and inflammation or convey protection. For this reason, Stage 3 RIC also failed to enhance survival of pups, unlike Stage 1 or 2 RIC which are administered at early stages of the disease to confer protection. Hence, it is unlikely that inflammation is the primary target of RIC for protecting the intestine.
- In the newly added experiments in our manuscript, we have also investigated the effects of nonspecific vasodilators, papaverine and captopril, in NEC pups. These drugs directly affect the regulation of blood flow, with no reports suggesting an effect on inflammation⁹. By demonstrating that administration of these vasodilators alone is sufficient to convey the same protective effects of RIC and improve the outcome of NEC, we provide further support that reduced inflammation is only a secondary outcome of RIC.

We appreciate the evidence provided by previous studies in the literature that RIC reduces inflammation. However, we hope that with our additional experiments, we are able to convince the reviewer that in this case, mitigation of inflammation is secondary to the preservation of intestinal blood flow in the mechanism of action of RIC for improving NEC outcome.

4. I remain very concerned by the data in Figure 1 that the early RIC occurs actually after the first time point in the late RIC group, so the data regarding early vs. late is uninterpretable. If the authors now believe that both early and late RIC is effective, given the fact that longer durations of RIC would be expected to have greater effects on H₂S or NO production, I'm even more concerned by the results.

We appreciate the reviewer's concern regarding the selected time points for the administration of RIC. Please allow us to address this confusion by illustrating a few points:

- In our revised manuscript, we updated the terminology of early and late RIC to Stage 1 and Stage 2 RIC, respectively. Please allow us to adhere to this terminology as it avoids further confusion.
- As explained in the Methods section of the manuscript, the time points chosen for Stage 1, Stage 2, and Stage 3 RIC are defined *only* with respect to the timeline of NEC induction. Our NEC induction protocol begins at P5 and ends at P9. Stage 1 RIC describes the first episode of RIC administered at the very early stage of disease induction, on P5. Stage 2 RIC describes the first episode of RIC administered at a later stage of disease induction, on P6. Stage 3 RIC describes the first episode of RIC administered at an even later stage of disease induction, on P7. Therefore, the terminology of Stage 1 vs. Stage 2 vs. Stage 3 RIC is used to distinguish between RIC administered on the first day of NEC induction, RIC administered on the second day of NEC induction, and RIC administered on the third day of NEC induction respectively.

The reason for confusion is the overlap between the timepoints of the three stages of RIC. However, as explained above, what defines and differentiates Stage 1 vs. Stage 2 vs. Stage 3 RIC is the first timepoint at which RIC is administered to NEC pups in the NEC induction timeline starting at P5 and ending at P9 (as illustrated in the Figure in this letter).

- Previous reports suggest that in addition to short-lasting protective effects that are conferred by RIC immediately after it is administered, RIC also activates a time window of protection which occurs 12-24 hours after it is administered, which lasts for 48-72 hours^{10,11}. This is why for each Stage of RIC (1, 2, 3) we administered RIC in two episodes and on two non-consecutive days during the 5-day period of NEC induction; to ensure that the second window of protection activated by RIC as suggested by the literature remains in effect until the day of sacrifice. By this logic:
 - Stage 1 RIC involves a first episode of RIC given on the first day of NEC induction (P5), followed by a second episode of RIC given 48 hours later, on the third day of NEC induction (P7).
 - Stage 2 RIC involves a first episode of RIC given on the second day of NEC induction (P6), followed by a second episode of RIC given 48 hours later, on the fourth day of NEC induction (P8).
 - Stage 3 RIC involves a first episode of RIC given on the third day of NEC induction (P7), followed by a second episode of RIC given 48 hours later, on the final day of NEC induction (P9).
- The importance of the chosen timepoints for RIC as explained above underlies its implication in the clinical setting. The prospect that we have for application of RIC in the clinical setting is to implement it for treatment of three groups of neonates with NEC:

- Neonates with Stage I NEC which present with suspected or minimal intestinal damage¹²⁻¹⁴
- Neonates with Stage II NEC, which present with moderate intestinal damage¹²⁻¹⁴
- Neonates with Stage III NEC, which present with severe intestinal damage¹²⁻¹⁴

In all of these patient groups, one single episode of RIC will not be effective for preventing the progressing of the disease and conferring protection. This stems from the explanation given above regarding the different time windows of protection activated by RIC. Hence, in the clinical setting, RIC will have to be administered on multiple days.

In our study design, Stage 1, 2 and 3 RIC simulate human Stage I NEC < Stage II NEC, and Stage III NEC respectively. Including these different timepoints is important as we have shown that Stage 3 RIC, which is administered at a very late stage of NEC development when significant intestinal injury has been established, is not effective in reversing the damage of NEC. Hence, we believe that from a clinical perspective, RIC can be implemented as a therapeutic strategy only to prevent progression of early NEC to more advanced stages of the disease requiring surgical intervention.

- Finally, while there is evidence that repeated episodes of RIC may promote beneficial effects¹⁵, according to our knowledge, “longer durations of RIC” have not been proven to be beneficial.

5. I remain concerned by the number of repeats and whether the data in Figures 2-3 and also 5-7 are biological vs. technical vs. experimental repeats.

We appreciate the reviewer’s comments regarding the number of repeats in our experiments. The *minimum* number of biological repeats are listed in the following. For the precise number of biological repeats for each experiment, please refer to the Figure Captions in the manuscript.

- BF control (n=10)
- NEC (n=10)
- NEC+Stage 1 RIC (n=10)
- NEC+Stage 2 RIC (n=10)
- NEC+Stage 3 RIC (n=9)
- *eNOS* BF (n=5)
- *eNOS* NEC (n=5)
- *eNOS* NEC+Stage 1 RIC (n=6)
- *eNOS* NEC+Stage 2 RIC (n=6)
- NEC+RIC+Chemical inhibitors of NO/H₂S (n=8 per group)
- NEC+NaHS (n=6)
- NEC+NaHS+Chemical inhibitors of H₂S (n=6)
- Regarding technical repeats, all experiments were performed using at least 3 repeats per sample.

- Regarding experimental repeats, all experiments include mice taken from various litters.

6. why is all RT-PCR data shown in aggregate as bars and not scatter plots? This is concerning given the high error bars.

We can modify the data in the manuscript as requested to display our qRT-PCR data using scatter plots.

Reviewer #2 (Remarks to the Author):

The authors should be commended for providing extensive new data, however some of the data does not address the initially raised concerns regarding mechanism of RIC effect, and thus overall enthusiasm for the manuscript is only modestly improved.

Top of page 15: The authors treat with H₂S synthesis inhibitors and show lack of protection by RIC against NEC. They conclude that the RIC effect is due to changes in intestinal perfusion. This is not an acceptable conclusion from these data. It is highly possible that H₂S is acting in some other way (than dilation) to mediate its beneficial effect since H₂S has a host of other biochemical and physiological effects. Flow changes may be secondary, not causative. The problem is that the experiments necessary to prove that it changes in flow are responsible, are difficult to do. One would need to add H₂S or nitric oxide and include a vasoconstrictor to prevent changes in flow to show that the beneficial effect of these agents on NEC was abrogated. In addition use of a nonspecific vasodilator such as Papaverine should be used to show that improved perfusion is sufficient to inhibit the effects of NEC. Short of such data (that dissociate changes in flow and other signaling actions of H₂S and NO) the authors cannot conclude that changes in flow are mechanistically responsible. The problem with using inhibitors of H₂S formation or of NO production (LNAME or eNOS KO) is that you end up blocking both the vasomotor effects and the many other biochemical signaling effects of these molecules, thus you cannot distinguish which is responsible for improvement in NEC via RPC.

We appreciate the comments from the reviewer. The constructive suggestions allowed us to conduct additional experiments to improve our manuscript and provide further support for our hypothesis.

- To prove that changes in intestinal blood flow are sufficient to improve the outcome of NEC, we followed the reviewer's suggestions to investigate the effects of nonspecific vasodilators. In Figure S7a-c, we demonstrate that administration of papaverine and captopril improves intestinal injury, reduces inflammation, and enhances survival. Using these nonspecific vasodilators, we have dissociated the changes in flow from other biochemical and physiological effects of NO and H₂S. Our current data demonstrates that preservation of intestinal microcirculation is responsible for conferring the RIC-mediated protection and reversing the intestinal damage of NEC.

- Additionally, we have investigated the effects of methoxamine, an intestinal vasoconstrictor² in the presence of RIC. As shown in Figure S7d, methoxamine abolished the beneficial effects of RIC in prolonging survival, which is the most crucial aspect of RIC-mediated protection in NEC from a clinical perspective. This data provides further support that vasodilation is essential to the ability of RIC to mitigate the intestinal damage of NEC and improve disease outcome.

We hope that with these additional experiments, we are able to convince the reviewer of the mechanism of action of RIC being targeted primarily towards the intestinal microcirculation.

Reviewer #3 (Remarks to the Author):

This report studies RIC in a murine model of neonatal enterocolitis. RIC is tested "early" (pre RIC days 5 and 7) and "late" (Days 6 and 8; per and post RIC) RIC is remarkably effective and the mechanism involves improved microperfusion. Improved perfusion with RIC has been shown previously in both brain and coronary circulations as the authors reference. The methods are elegant. This work is highly translatable as the authors add to the evidence that RIC is safe and effective

Some questions

1.) Was RIC applied on one or both hindlimbs? Not clear in lines 305-8

We appreciate the comment from the reviewer. The RIC stimulus was always given to one hind limb and the same hind limb was used for all pups throughout the various experiments. We have clarified this in the manuscript.

2.) How was this regimen of RIC chosen? They were certainly effective but how were the regimens arrived at?

The phenomenon of ischemic pre-conditioning was first described in the canine heart wherein four 5 min circumflex coronary occlusions, each separated by 5 min of reperfusion, dramatically reduced myocardial infarction size¹⁶. Przyklenk et al. also showed that brief myocardial ischemia by four cycles of 5 min coronary artery occlusion protected local and remote myocardium from sustained 1 h cardiac ischemia reperfusion injury¹⁷. Experimental and clinical evidence also suggests that RIC activates at least two distinct time frames of protection against ischemia reperfusion injury of the brain and heart^{10,11}. The first window of protection occurs immediately after the RIC stimulus and lasts for 2 hours, and involves changes in ion channel permeability, protein phosphorylation, and release of several signaling mediators¹⁰. The second window of protection follows 12-24 hours after the RIC stimulus and lasts 48-72 hours, involving modulation of inflammatory response, improved endothelial function, and activation of gene expression¹¹. Moreover, previous authors have reported that compared with a single episode of

remote ischemic conditioning, repeated episodes were more protective in reducing inflammation in the ischemic myocardium¹⁵. Hence, the previously used protocol of four cycles of 5 min occlusion followed by 5 min reperfusion was chosen for RIC in our model. We selected two time periods of RIC (Stage 1 RIC at P5 and P7, and Stage 2 RIC at P6 and P8) which simulate two initial stages of NEC, Stage IIA and Stage IIB respectively^{13,14}. The efficacy of RIC in both time periods is important for the translational application of this experimental observation. We added an explanation in the manuscript.

3.) The authors studied the H₂S system. The NO and NOS system has also been implicated in the mechanism of action of RIC and are important in perfusion. Were NOS 3 inhibitors tested or NOS3 KO mice? The authors might comment on this

Previous authors have reported that H₂S improves mesenteric perfusion and intestinal injury in experimental NEC via an *eNOS*-dependent mechanism¹, suggesting that NO-mediated regulation of intestinal microcirculation occurs downstream of H₂S¹⁸. To further illustrate the importance of NO in RIC-mediated vasodilation during experimental NEC, we performed additional experiments using *eNOS* knockout mice. We demonstrated that RIC in these mice had no beneficial effect on NEC intestinal injury (morphology and inflammation) and did not improve survival. The baseline perfusion of various organs, including the intestine, was tenuous due to lack of *eNOS* and the measurement of intestinal wall perfusion during experimental NEC and RIC was not reliable (Supplementary Figure S3). To overcome this difficulty, we explored the role of NO synthase inhibitor L-NAME on the effects RIC in experimental NEC. Our findings demonstrated that the RIC-mediated improvement in the perfusion of the whole intestinal wall and of the submucosal layer was abolished upon treatment with L-NAME. Consistent with *eNOS* knock out mice, inhibition of NO by L-NAME eliminated the RIC beneficial effects of reducing NEC-related intestinal injury and inflammation. The findings from these additional experiments can be found in the results, and in Fig. 2g-h, Fig. 6, Fig. 7, and supplementary Fig. S1c, and Fig. S6.

Final considerations

We would like to thank the editors for giving us the chance to clarify the comments and concerns of the reviewers. This is a valuable opportunity and we hope that we have been able to provide further clarification for the remaining concerns of the reviewers.

The main concern of Reviewer #1 seemed to be related to whether the changes in intestinal perfusion conferred by RIC precedes changes in intestinal inflammation. We hope that our explanations can convince the reviewer that while RIC has been shown to primarily target inflammatory cascades in previous studies, in our study, the primary effect of RIC is preservation of intestinal microcirculation, followed by decreased injury and inflammation in the intestine, enhanced survival, and hence improvement of NEC outcome.

We were happy that after our modifications and added experiments in the revised manuscript, Reviewer #2 agreed that RIC confers its protective effects in the NEC intestine by improving perfusion across the entire intestinal wall, as well as in the submucosal arterioles and within the villi. In our resubmitted manuscript, we followed the advice of the reviewer and performed additional experiments to further investigate the role of vasodilation in the RIC-mediated effects on intestinal blood flow dynamics. We hope that by investigating the effects of nonspecific vasodilator and vasoconstrictor agents, we have been able to show that vasodilation alone is sufficient to confer the protection of RIC in the NEC intestine.

Finally, we assume that we have been successful in addressing all of the comments and concerns of Reviewer #3, who had reviewed our original manuscript, as no further comments were provided by this reviewer.

Thank you very much for your consideration of our manuscript for publication in *Nature Communications*.

References

1. Wang, G. L., Jiang, B. H., Rue, E. A. & Semenza, G. L. Hypoxia-inducible factor 1 is a basic-helix-loop-helix-PAS heterodimer regulated by cellular O₂ tension. *Proc Natl Acad Sci U S A* **92**, 5510-5514, doi:10.1073/pnas.92.12.5510 (1995).
2. Wang, G. L., Jiang, B. H., Rue, E. A. & Semenza, G. L. Hypoxia-inducible factor 1 is a basic-helix-loop-helix-PAS heterodimer regulated by cellular O₂ tension. *Proc Natl Acad Sci U S A* **92**, 5510-5514, doi:10.1073/pnas.92.12.5510 (1995).
3. Chen, Y. *et al.* Formula feeding and immature gut microcirculation promote intestinal hypoxia, leading to necrotizing enterocolitis. *Dis Model Mech* **12**, doi:10.1242/dmm.040998 (2019).
4. Hsueh, W. *et al.* Neonatal necrotizing enterocolitis: clinical considerations and pathogenetic concepts. *Pediatr Dev Pathol* **6**, 6-23, doi:10.1007/s10024-002-0602-z (2003).
5. Downard, C. D. *et al.* Altered intestinal microcirculation is the critical event in the development of necrotizing enterocolitis. *J Pediatr Surg* **46**, 1023-1028, doi:10.1016/j.jpedsurg.2011.03.023 (2011).
6. Yazji, I. *et al.* Endothelial TLR4 activation impairs intestinal microcirculatory perfusion in necrotizing enterocolitis via eNOS-NO-nitrite signaling. *Proc Natl Acad Sci U S A* **110**, 9451-9456, doi:10.1073/pnas.1219997110 (2013).
7. Yu, X., Radulescu, A., Zorko, N. & Besner, G. E. Heparin-binding EGF-like growth factor increases intestinal microvascular blood flow in necrotizing enterocolitis. *Gastroenterology* **137**, 221-230, doi:10.1053/j.gastro.2009.03.060 (2009).
8. Dyson, R. M. *et al.* A role for H₂S in the microcirculation of newborns: the major metabolite of H₂S (thiosulphate) is increased in preterm infants. *PLoS One* **9**, e105085, doi:10.1371/journal.pone.0105085 (2014).
9. Zani, A. *et al.* Captopril reduces the severity of bowel damage in a neonatal rat model of necrotizing enterocolitis. *J Pediatr Surg* **43**, 308-314, doi:10.1016/j.jpedsurg.2007.10.022 (2008).
10. Ren, C., Gao, X., Steinberg, G. K. & Zhao, H. Limb remote-preconditioning protects against focal ischemia in rats and contradicts the dogma of therapeutic time windows for preconditioning. *Neuroscience* **151**, 1099-1103, doi:10.1016/j.neuroscience.2007.11.056 (2008).
11. Kuzuya, T. *et al.* Delayed effects of sublethal ischemia on the acquisition of tolerance to ischemia. *Circulation research* **72**, 1293-1299, doi:10.1161/01.res.72.6.1293 (1993).
12. Lee, J. S. & Polin, R. A. Treatment and prevention of necrotizing enterocolitis. *Seminars in neonatology : SN* **8**, 449-459, doi:10.1016/S1084-2756(03)00123-4 (2003).
13. Walsh, M. C. & Kliegman, R. M. Necrotizing enterocolitis: treatment based on staging criteria. *Pediatric clinics of North America* **33**, 179-201, doi:10.1016/s0031-3955(16)34975-6 (1986).
14. Bell, M. J. *et al.* Neonatal necrotizing enterocolitis. Therapeutic decisions based upon clinical staging. *Annals of surgery* **187**, 1-7, doi:10.1097/00000658-197801000-00001 (1978).
15. Jiang, Q. *et al.* Systemic redistribution of the intramyocardially injected mesenchymal stem cells by repeated remote ischaemic post-conditioning. *Journal of cellular and molecular medicine* **22**, 417-428, doi:10.1111/jcmm.13331 (2018).

16. Murry, C. E., Jennings, R. B. & Reimer, K. A. Preconditioning with ischemia: a delay of lethal cell injury in ischemic myocardium. *Circulation* **74**, 1124-1136, doi:10.1161/01.cir.74.5.1124 (1986).
17. Przyklenk, K., Bauer, B., Ovize, M., Kloner, R. A. & Whittaker, P. Regional ischemic 'preconditioning' protects remote virgin myocardium from subsequent sustained coronary occlusion. *Circulation* **87**, 893-899, doi:10.1161/01.cir.87.3.893 (1993).
18. Drucker, N. A., Jensen, A. R., Te Winkel, J. P. & Markel, T. A. Hydrogen Sulfide Donor GYY4137 Acts Through Endothelial Nitric Oxide to Protect Intestine in Murine Models of Necrotizing Enterocolitis and Intestinal Ischemia. *The Journal of surgical research* **234**, 294-302, doi:10.1016/j.jss.2018.08.048 (2019).

Reviewers' Comments:

Reviewer #1:

Remarks to the Author:

The authors are to be congratulated for taking the comments extremely seriously, and performing a thorough and methodical revised manuscript. Within the limits of in vivo experiments and with the addition of the human tissue work, my concerns have been addressed. The work is vastly improved, and more mechanistic.

Reviewer #2:

Remarks to the Author:

The authors were responsive to continued concerns. Some of the additional data provided helps in the interpretation of the mechanism by which RIC is protective. For example a key experiment is the demonstration that papaverine-induced increases in blood flow mimic the effect of RIC, and that methoxamine, an alpha1 adrenergic agonist blocks the benefit of RIC.

However the authors continue to overinterpret their data in other ways. 1. They argue that RIC does not alter inflammation as a means for protection since initiating RIC during the time of peak inflammation following induction of NEC, has no benefit, but it does when administered early on. Inflammation is a cascading event and it is not surprising that RIC is preventative and not therapeutic in this regard. It tells us little about causality.

The additional data in eNOS null mice does nothing to help solve the question about flow as a mechanism of RIC. NO has anti-inflammatory actions that could be playing a preventative role, independent of vasomotor effects of NO, which although established, can rarely compete with metabolic forms of dilation which are much more profound in determining perfusion. The authors still state that RIC works through vasodilation via NO when in fact the data to support this are not provided.

Reviewer #4:

Remarks to the Author:

I was asked to comment on the response to reviewer 3. The response is satisfactory. The authors specified that RIC was applied to the left limb. However, I believe it would be useful for readers to know what the tourniquet was made of (tubing? cord?), precisely how it was applied to a tiny mouse pup, and what attempts, if any, were made to try to achieve a consistent pressure, without damaging the limb. This will be essential for efforts to replicate this novel experimental model.

I have some additional comments on the manuscript:

The authors make the strong statement that "while RIC has been shown to primarily target inflammatory cascades in previous studies, in our study, the primary effect of RIC is preservation of intestinal microcirculation." The only reported measure of inflammation that I can see in this study is Il-6 levels, and in fact this measure was significantly reduced by both RIC-S1 and RIC-S2 (fig S1b). While it is difficult to distinguish direct effects from secondary effects, they cannot dismiss the possibility that RIC directly decreased inflammation and that this provided some benefit in NEC in addition to vascular effects. This possibility should be included in the discussion.

Overall, I agree with other reviewers that the authors conclusion "The mechanism of action of RIC is increasing intestinal perfusion through vasodilation mediated by nitric oxide and hydrogen sulfide." is too strong, and should be amended to "The mechanism of action of RIC involved an increase in intestinal perfusion through vasodilation mediated by nitric oxide and hydrogen sulfide."

The lack of protection in eNOS mice is compelling, but does not prove that the effect is via the

endothelium since eNOS is expressed in other cell types in addition to endothelium. Indeed, eNOS is expressed in macrophages, and may therefore influence inflammation. J Biol Chem. 2003 Jul 18;278(29):26480-7 Furthermore it does not prove that the effect is via vasodilation (although this is likely to contribute), since endothelial-derived NO also has effects on hemostasis and circulating inflammatory cells.

The observation that in the presence of methoxamine (a vasoconstrictor), the protective effects of RIC are lost, does not prove that RIC is protective via vasorelaxation. It is to be expected that a vasoconstrictor would worsen NEC phenotype, even in the presence of protective agent (whether it is working via vasculature or not). Notably, there was no group with NEC+meth to examine this possibility.

The above points should be discussed, or at least, some room given in the discussion to the possibility that the inflammatory suppressive effects on RIC that were seen might contribute to the outcome benefits.

Response to Reviewer's Comments

Remote ischemic conditioning counteracts the intestinal damage of necrotizing enterocolitis by improving intestinal microcirculation

We would like to thank the editor and the reviewers for their careful review of our responses to the previous comments from the reviewers, and for the thoughtful and constructive comments. Our responses to the reviewers' comments are in blue in this letter. The corresponding modifications in the manuscript are highlighted in yellow.

Reviewer #1 (Remarks to the Author):

The authors are to be congratulated for taking the comments extremely seriously, and performing a thorough and methodical revised manuscript. Within the limits of in vivo experiments and with the addition of the human tissue work, my concerns have been addressed. The work is vastly improved, and more mechanistic.

We would like to thank Reviewer #1 for their comments. We are happy that we have been able to address all this reviewer's concerns. We thank the reviewer for their suggestions and comments which greatly helped us improve our manuscript.

Reviewer #2 (Remarks to the Author):

The authors were responsive to continued concerns. Some of the additional data provided helps in the interpretation of the mechanism by which RIC is protective. For example a key experiment is the demonstration that papaverine-induced increases in blood flow mimic the effect of RIC, and that methoxamine, an alpha1 adrenergic agonist blocks the benefit of RIC.

However the authors continue to overinterpret their data in other ways. 1. They argue that RIC does not alter inflammation as a means for protection since initiating RIC during the time of peak inflammation following induction of NEC, has no benefit, but it does when administered early on. Inflammation is a cascading event and it is not surprising that RIC is preventative and not therapeutic in this regard. It tells us little about causality.

We thank the reviewer for their comments regarding the revised manuscript, and we appreciate the reviewer's remaining concerns.

We understand the reviewer's comments regarding the causality of RIC. We agree that we cannot rule out the activation and involvement of an inflammatory cascade downstream of RIC. We have modified the interpretation of our data regarding the lack of protection by RIC at later stages of NEC induction (Please see lines 319-329, 404-408, and Fig. 9 lines 928).

The additional data in eNOS null mice does nothing to help solve the question about flow as a mechanism of RIC. NO has anti-inflammatory actions that could be playing a preventative role, independent of vasomotor effects of NO, which although established, can rarely compete with metabolic forms of dilation which are much more profound in determining perfusion. The

authors still state that RIC works through vasodilation via NO when in fact the data to support this are not provided.

We also understand the reviewer's comments regarding the broader role of NO in the mechanism of action of RIC. We understand that we cannot rule the possibility that the role of NO in the RIC-mediated protection could be due to its the anti-inflammatory actions independent of the vasomotor effects of NO. We have added this explanation in our manuscript (Please see lines 319-329).

Reviewer #4 (Remarks to the Author):

I was asked to comment on the response to reviewer 3. The response is satisfactory. The authors specified that RIC was applied to the left limb. However, I believe it would be useful for readers to know what the tourniquet was made of (tubing? cord?), precisely how it was applied to a tiny mouse pup, and what attempts, if any, were made to try to achieve a consistent pressure, without damaging the limb. This will be essential for efforts to replicate this novel experimental model.

We would like to thank this reviewer for their review of our manuscript. We appreciate all the reviewer's comments and concerns.

The tourniquet used for administration of RIC was a surgical rubber vessel loop of 1 mm diameter. RIC was induced by placing the tourniquet around the base of the left hind limb.

Due to technical limitations for mouse pups of this size (postnatal day 5-9 and body weight 3-5 grams), we were not able to obtain mechanical measurements of pressure such as the arterial pulse wave transit time. However, the hind limb occlusion and reperfusion in each cycle of RIC were verified by monitoring the change in color of the limb. A change in color was consistently achieved in less than 30 seconds. We have added the above information to the "Methods" section of the manuscript (Please see lines 445-452).

To ensure that the hind limb occlusion does not damage the limb, we performed three different functional motor tests on control pups receiving the RIC stimulus. We showed that RIC did not cause any injuries and produced no deficits in the motor function and strength of the limb. Please see Fig. S9a-c and Videos S17-19.

I have some additional comments on the manuscript:

The authors make the strong statement that "while RIC has been shown to primarily target inflammatory cascades in previous studies, in our study, the primary effect of RIC is preservation of intestinal microcirculation." The only reported measure of inflammation that I can see in this study is Il-6 levels, and in fact this measure was significantly reduced by both RIC-S1 and RIC-S2 (fig S1b). While it is difficult to distinguish direct effects from secondary effects, they cannot dismiss the possibility that RIC directly decreased inflammation and that this provided some benefit in NEC in addition to vascular effects. This possibility should be included in the discussion.

Overall, I agree with other reviewers that the authors conclusion "The mechanism of action of

RIC is increasing intestinal perfusion through vasodilation mediated by nitric oxide and hydrogen sulfide.” is too strong, and should be amended to “The mechanism of action of RIC involved an increase in intestinal perfusion through vasodilation mediated by nitric oxide and hydrogen sulfide.”

We agree with the reviewer’s important comment regarding the possibility of anti-inflammatory effects of RIC. We are convinced that we cannot dismiss the possibility that RIC directly decreases inflammation in its mechanism of action, along with modulation of intestinal blood flow dynamics. We have included this possibility in the “Discussion” section of our manuscript (Please see lines 319-329).

We are therefore convinced that the conclusion of our manuscript must be amended to acknowledge the possibility that RIC also modulates the inflammatory cascade in its mechanism of protection. We have modified our conclusion in the manuscript accordingly (Please see lines 50, 404-408, and Fig. 9 lines 928).

The lack of protection in eNOS mice is compelling, but does not prove that the effect is via the endothelium since eNOS is expressed in other cell types in addition to endothelium. Indeed, eNOS is expressed in macrophages, and may therefore influence inflammation. J Biol Chem. 2003 Jul 18;278(29):26480-7 Furthermore it does not prove that the effect is via vasodilation (although this is likely to contribute), since endothelial-derived NO also has effects on hemostasis and circulating inflammatory cells.

We thank the reviewer for drawing our attention to the important work of Connelly L, et al on macrophage *eNOS* and its role in the initiation of an inflammatory response. We agree that we cannot rule out effects other than vasodilation by endothelial-derived NO such as its effects on homeostasis and circulating inflammatory cells. We have included a discussion of this paper in the “Discussion” section of our manuscript (Please see lines 319-329, and reference #44, 45).

The observation that in the presence of methoxamine (a vasoconstrictor), the protective effects of RIC are lost, does not prove that RIC is protective via vasorelaxation. It is to be expected that a vasoconstrictor would worsen NEC phenotype, even in the presence of protective agent (whether it is working via vasculature or not). Notably, there was no group with NEC+meth to examine this possibility.

We agree with the reviewer that administration of methoxamine is expected to worsen NEC phenotype, even in the presence of RIC. Indeed, we have demonstrated this in Fig. S7d. Administration of methoxamine, in the presence of RIC, produced significantly worse survival rates than NEC alone. Due to this observation, we did not include a NEC+methoxamine experimental group as we anticipated severe mortality for this group (Please see lines 334-336).

The above points should be discussed, or at least, some room given in the discussion to the possibility that the inflammatory suppressive effects on RIC that were seen might contribute to the outcome benefits.

As mentioned above, we acknowledge that we cannot rule out the possibility that RIC may induce direct effects on inflammatory pathways, which contribute to the overall protection by RIC. We have added these discussions to the manuscript (Please see lines 50, 404-408, and Fig. 9 legend lines 928).

We thank all the reviewers again for their careful review of our revised manuscript. We appreciate the constructive comments and suggestions. We agree on the need to tone down and revise the conclusions of the manuscript to acknowledge that the potential role for inflammatory mechanisms of RIC cannot be excluded. We hope to have successfully addressed all the concerns with the new modifications to our discussion of the mechanism of action of RIC.

Reviewers' Comments:

Reviewer #2:

Remarks to the Author:

Each concern has been addressed and the conclusions now reflect the data presented.

Reviewer #4:

Remarks to the Author:

The authors have added appropriate discussion which addresses the concerns I had raised satisfactorily.